# Behavioral evidence for nested central pattern generator control of *Drosophila* grooming

**Primoz Ravbar\*, Neil Zhang, Julie H Simpson\***

Molecular Cellular and Developmental Biology and Neuroscience Research Institute, University of California, Santa Barbara, Santa Barbara, United States

**Abstract** Central pattern generators (CPGs) are neurons or neural circuits that produce periodic output without requiring patterned input. More complex behaviors can be assembled from simpler subroutines, and nested CPGs have been proposed to coordinate their repetitive elements, organizing control over different time scales. Here, we use behavioral experiments to establish that *Drosophila* grooming may be controlled by nested CPGs. On a short time scale (5–7 Hz, ~ 200 ms/movement), flies clean with periodic leg sweeps and rubs. More surprisingly, transitions between bouts of head sweeping and leg rubbing are also periodic on a longer time scale (0.3–0.6 Hz, ~2 s/bout). We examine grooming at a range of temperatures to show that the frequencies of both oscillations increase—a hallmark of CPG control—and also that rhythms at the two time scales increase at the same rate, indicating that the nested CPGs may be linked. This relationship holds when sensory drive is held constant using optogenetic activation, but oscillations can decouple in spontaneously grooming flies, showing that alternative control modes are possible. Loss of sensory feedback does not disrupt periodicity but slow down the longer time scale alternation. Nested CPGs simplify the generation of complex but repetitive behaviors, and identifying them in *Drosophila* grooming presents an opportunity to map the neural circuits that constitute them.

**\*For correspondence:**
primoz.ravbar@gmail.com (PR);
jhsimpson@ucsb.edu (JHS)

**Competing interest:** The authors declare that no competing interests exist.

## Editor's evaluation

This study presents an intriguing example of natural rhythmic leg movement oscillations at a fast time scale embedded within a slower rhythmic behavioral oscillation and the similar effects of temperature on the rates of the two rhythms. The data and the writing are both exceptionally clear and advance our understanding,

## Introduction

Animals combine simpler movements into complex routines, forming behaviors with organization across multiple time scales. We have observed lobsters moving antennal segments at different speeds, and marveled at pianists playing Rachmaninoff moving their fingers blindingly fast, hands at a slower pace, while arms and body are swinging, all nearly perfectly orchestrated. We wondered how a complex behavior can be assembled from simpler movements in such a harmonious manner.

Central pattern generators (CPGs) are neural circuits that produce rhythmic motor outputs in response to a trigger without requiring ongoing descending drive or patterned sensory inputs; CPGs control short stereotypic actions in cat walking, crayfish swimming, locust flight, leech heartbeat, and the stomatogastric and pyloric rhythms of crustaceans (reviewed in *Berkowitz, 2019*; *Harris-Warrick and Ramirez, 2017* ; *Grillner, 2006*; *Marder and Calabrese, 1996*; *Mulloney and Smarandache, 2010*; *Selverston, 2010*). However, CPGs may also contribute to control of more complex behaviors.

When the movements that compose a behavior repeat, it is inefficient to initiate each step with a separate decision. Automating the sequence by calling its actions in series produces reliable execution. Increasingly complex sequences can be assembled from shorter elements, suggesting hierarchical control. When repetitive subroutines are themselves composed of simpler periodic movements, they may be controlled by nested CPGs, hierarchically organized so that a 'high-level' slow CPG controls the behavior on coarse scale and a 'lower-level' fast CPG adds the fine structure (*Berkowitz, 2019*). In other words, the slow CPG controls alternations *between* subroutines and the fast CPG controls alternations *within* these subroutines. Various combinations of coarse and fine oscillators could produce behaviors of arbitrary complexity while still keeping them well-timed, stereotyped, and coherent. Bird song, for example, contains sound syllables executed in sequences. The syllables are short repeating elements, and sequences of syllables make phrases or words that also repeat, creating structure over several time scales. Ingenious local cooling experiments of specific brain regions cause the whole song to slow down, indicating that it is governed by central pattern generating circuits (*Long and Fee, 2008*). Temporally nested, hierarchical patterns of neural activity have also recently been found during locomotion in *Caenorhabditis elegans* (*Kaplan et al., 2020*).

Here, we show that *Drosophila* grooming behavior contains periodic elements over several time scales of which we explore two: a fast repeat of individual leg movements (including head sweeps and front leg rubs) and a slow alternation between bouts of head cleaning and front leg rubbing. We demonstrate that both of these repeated elements show evidence of CPG control, and that the two rhythms are usually coordinated, establishing fly grooming as a model system for understanding the circuit architecture of nested CPGs.

## Results

### Two time scales of grooming are periodic

When flies are covered in dust, they initially groom anterior body parts using their front legs (*Seeds et al., 2014*). They alternate between bouts of head sweeps, where the legs move synchronously, and bouts of leg rubbing, where the legs move in opposition to each other, scraping the dust off. These movements are shown schematically in *Figure 1A*: the purple and orange arrows indicate synchronous in-phase head sweeps and opposing out-of-phase leg rubs, respectively; the thicker light blue arrows show alternation between these two leg coordination modes. Bouts of head sweeping (h) are indicated in purple and front leg rubbing (f) in orange on the ethogram (record of behavior actions over time) shown in *Figure 1B*.

The individual leg sweeps and rubs are stereotyped: these movements are recognizable by human observers or machine vision algorithms (*Mathis et al., 2018*; *Ravbar et al., 2019*), and they represent the short time scale we consider here. We first count individual leg movements from raw videos as they are processed for our Automatic Behavior Recognition System (ABRS) pipeline (see Materials and methods and *Figure 1—figure supplement 1*) and compute their frequencies. In *Figure 1C*, we show an example of frequencies of leg sweeps and rubs during the same one-minute period as *Figure 1B*. At 18°C, leg rubs and sweeps have a characteristic frequency ~6 Hz. This means that each leg movement takes approximately 150 ms to complete, which is consistent with our observations using higher resolution video recordings (see below).

Grooming is not a fixed action pattern and flies choose subroutines such as head sweeps or wing cleaning stochastically, but with different levels of prioritization (*Seeds et al., 2014*). During anterior grooming, flies typically make several head sweeps in a row, followed by repeated leg rubbing movements; we call these periods where a single type of action is repeated *bouts*. We use ABRS (*Ravbar et al., 2019*) to automatically classify different grooming actions and to identify the time points of maximum behavior identification confidence as the centers of bouts (orange and purple circles in the time series shown in *Figure 1D*). Head cleaning bouts and front leg rubbing bouts alternate: we define the time between two head cleaning bouts as a *hh-cycle* while the time between two consecutive front leg rubbing bouts is an *ff-cycle*. Either cycle will include both leg rubs and head sweeps, which is why we consider the average frequency of both types of leg movements in our analysis. These terms are illustrated in *Figure 1E*, and the cycles are the long time scale we investigate here.

We demonstrate the periodicity of head sweeps and leg rubs (short time scale) using autocorrelation analysis (*Figure 1F and G*). Although previous work revealed some syntactic organization at the

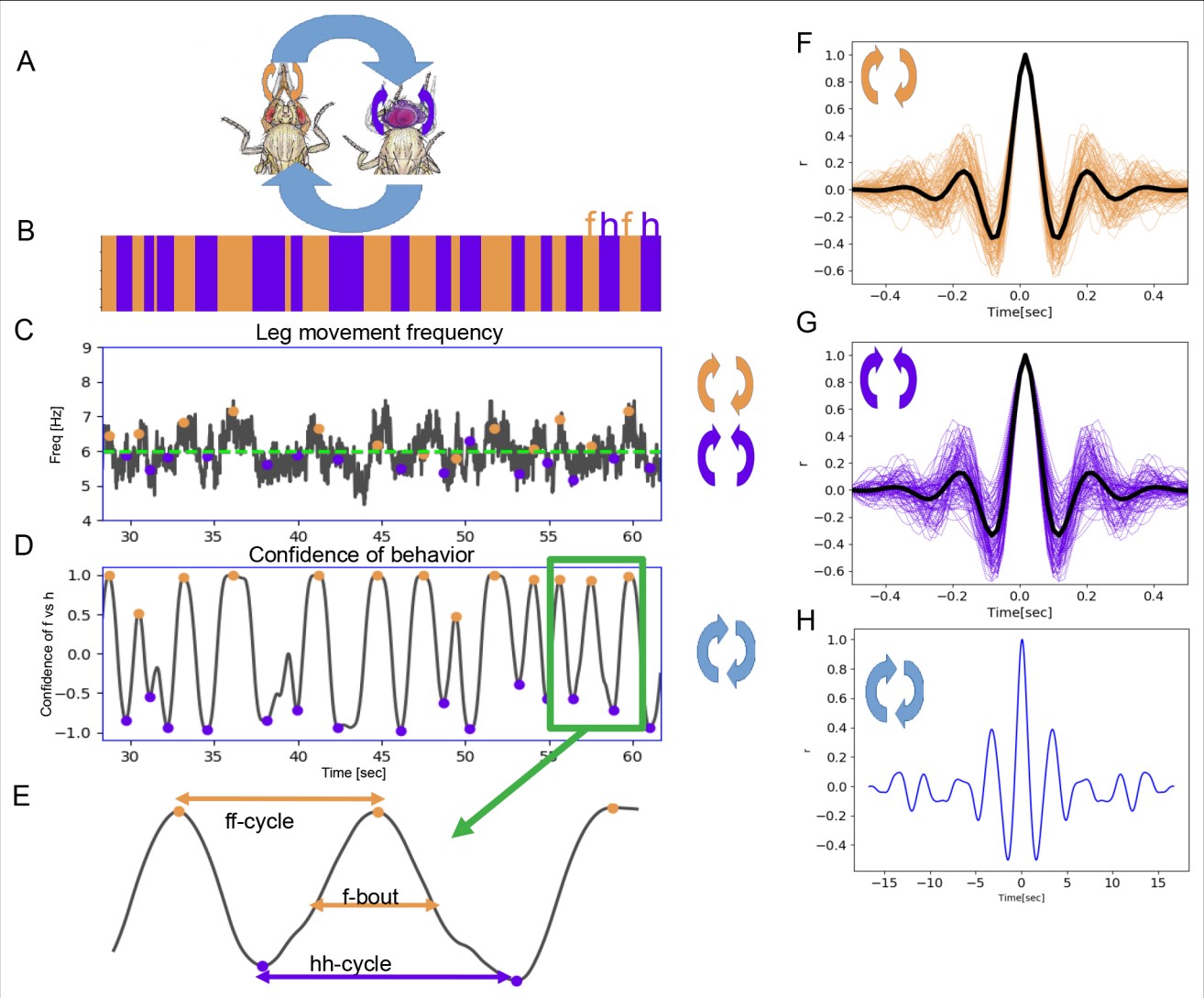

**Figure 1.** Two time scales of grooming behavior are periodic. (**A**) Schematic of anterior grooming behavior. The out-of-phase motion of leg rubbing is indicated by the orange arrows and in-phase head cleaning movements are indicated by the purple arrows (the short time scale). The blue arrows indicate alternations between the leg rubbing and head cleaning subroutines (the long time scale). (**B**) Ethogram showing alternations between bouts of front leg rubbing (f) and head cleaning (h) in dust-covered flies recorded at 18°C. (**C**) Example leg sweep and rub frequencies measured in the 30 s of anterior grooming behavior shown in the ethogram above. Purple and orange dots indicate front leg rubbing (f) and head cleaning (h) as detected by the Automatic Behavior Recognition System. (**D**) Bouts of front leg rubbing (f) or head cleaning (h) are identified by their probabilities (from the output of the Convolutional Neural Network). When we subtract the probability of h-bouts from that of f-bouts, we obtain the confidence of behavior curve shown here (see Materials and methods). Purple and orange dots indicate maxima and minima of confidence in behavior identification, corresponding to the centers of the f- and h-bouts, respectively. (**E**) Enlarged segment taken from (**D**) showing the definitions of ff-cycle and hh-cycle and f-/h-bouts. (**F**) Samples of autocorrelation functions (ACFs) computed over 3 min of movies when the fly was engaged in front leg rubbing or head sweeps (**G**). The thick black lines indicate the average of these samples, while thinner purple and orange lines represent each individual ACF contributing to this average; see Materials and methods for details. (**H**) ACF of the alternation of f-bout and h-bout from the example of ff-cycles shown in (**D**) also reveals periodic signal.

The online version of this article includes the following figure supplement(s) for figure 1:

**Figure supplement 1.** Methods of leg movements (sweeps or rubs) counting from video.

**Figure supplement 2.** Periodicity Index (PI) is used to measure the strength of periodicity and to separate periodic from non-periodic behaviors.

**Figure supplement 3.** Analysis of periodicity of both time scales for dust-stimulated flies, spontaneously grooming and optogenetically stimulated flies.

**Figure supplement 4.** Periodicity of both time scales is confirmed by an independent method of behavior analysis.

bout level—both identity and duration of current action influence the identity and duration of the next (*Mueller et al., 2019*)—we were surprised to find that the alternation of head cleaning and front leg rubbing bouts is also periodic. Autocorrelation analysis of the time series shown in *Figure 1H* illustrates signal at ~0.33 Hz, corresponding to approximately 2 s between the mid-point of consecutive bouts of front leg rubbing.

To quantify the strength of periodicity, we computed the ratio between the heights of the prominent peak nearest to zero lag and the central peak (*at* zero lag) of the autocorrelation function (ACF) (*Figure 1—figure supplement 2*); we call the ratio of these peak heights the Periodicity Index (PI). More periodic movements have high shoulder peaks: a perfect sine wave would have a PI of 1, while in more weakly periodic data, this index would approach 0. The weakest periodic grooming movements we observe have values ~0.2. Non-periodic signals have poorly defined shoulder peaks and fall below our threshold for peak detection; therefore, the PI is not computed for non-periodic behaviors (see Materials and methods). Using this metric, head sweeps and leg rubbing movements of dusted flies grooming at 18°C have a mean PI of 0.31 (s.d.=0.02) and 0.31 (s.d.=0.01), respectively; the alternation between bouts (the long time scale) at 18°C has a mean PI of 0.36 (s.d.=0.02). We also quantify the amount of periodic behavior as a ratio of time spent in periodic versus non-periodic movements, as shown in *Figure 1—figure supplement 2*. Using these ratios, we found that overall the amount of periodic behavior for the long time scale is 80% (s.d.=8%) at 18°C. For a comprehensive comparison of PI values and the overall amounts of periodic behaviors for the different conditions evaluated throughout this project see *Figure 1—figure supplement 3*. Illustrative examples of animals engaged in periodic or non-periodic behaviors, sampled from different groups studied, are shown in *Figure 6—figure supplement 2*.

We confirm the frequency and periodicity of leg movements during grooming using Deep Lab Cut (DLC) to analyze an independent video data set. DLC is a method developed for tracking of individual body parts (*Mathis et al., 2018*). *Figure 1—figure supplement 4* shows the changes of joint angles (resulting from leg movements) over 2 s of grooming behavior: at 18°C, the movement frequencies for the short time scale and long time scale are 4.5 Hz and 0.22 Hz, respectively, similar to our original measurements using ABRS. The PI values are ~0.34 and the prominence of shoulder peaks is above our threshold for periodic behaviors. The discovery that both short time scale and long time scale subroutines within grooming behavior show periodicity suggests the possibility that they may both be controlled by central pattern generating circuits.

## The period lengths of both time scales contract with increasing temperature

A key feature of CPGs is that they oscillate faster at higher temperatures (*Deliagina et al., 1983*; *Tang et al., 2012*). To determine whether temperature affects the periodicity of leg sweeps and rubs (short time scale) or the alternation between bouts of head sweeps and leg rubs (ff/hh-cycles; long time scale), we recorded the grooming behavior of dust-covered flies at a range of temperatures between 18°C and 30°C. Examples of ethograms from the extreme temperatures are shown in *Figure 2A*, and ethograms from the entire data set arranged from coolest to warmest temperature of 84 individual flies at seven temperatures recorded for 13 min each are displayed in *Figure 2B*.

Temperature increase causes faster individual leg movements (short time scale) (*Figure 2C–D*) and the frequency shows a linear increase from 5.7 Hz to 6.6 Hz ($R^2$=0.99, p<0.001; *Figure 2E*). This analysis combines sweeps and rubs, but when the different leg movements are considered separately, both show a similar increase with temperature (*Figure 2—figure supplement 1*).

The period of long time scale movements is also compressed by temperature. The ff/hh-cycle frequency increases from 0.42 Hz (s.d.=0.02) at 18°C to 0.49 Hz (s.d.=0.04) at 30°C, also in a linear manner ($R^2$=0.96, p<0.001; *Figure 2F–H*). The frequencies of hh-cycles show a very similar trend (*Figure 2—figure supplement 2*). Autocorrelations for the long time scale are more variable at higher temperatures, but the alternation remains periodic across the entire range of temperatures (*Figure 1—figure supplement 3F*, *Figure 2—figure supplement 3*). A full spectral analysis of the effect of temperature on the ACF shows complex structure over several time scales (*Figure 2—figure supplement 4*).

Increasing temperature shortens the cycle period of both the short time scale leg sweeps and rubs and the long time scale alternation between bouts of head cleaning and front leg rubbing. Next, we

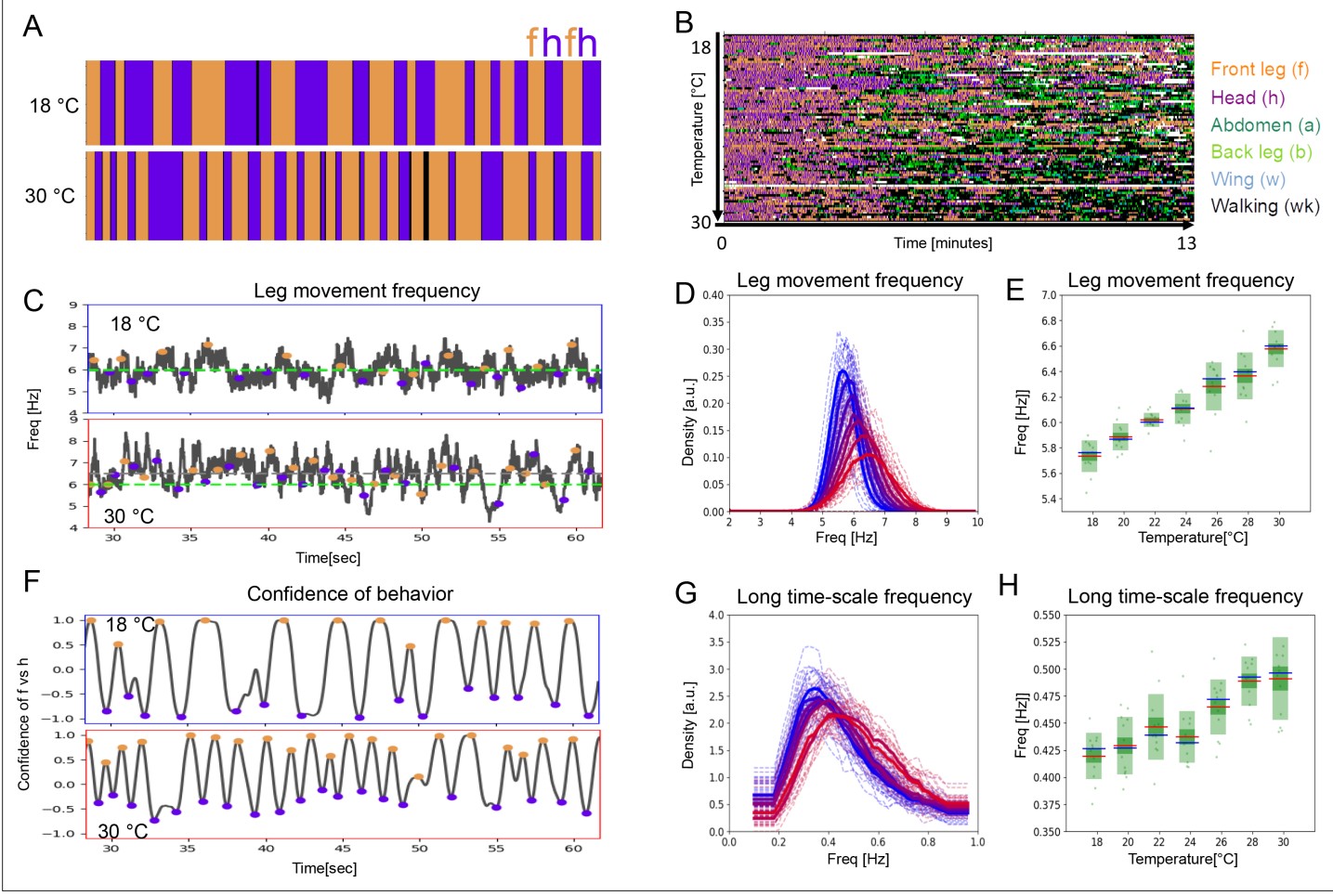

**Figure 2.** Period lengths of two time scales contract with increasing temperature in dust-stimulated flies. (**A**) Examples of ethograms recorded at 18°C (top) and 30°C (bottom). (**B**) Ethograms of 84 dust-stimulated flies recorded at different temperatures (18–30°C). Colors represent behaviors as indicated in the color legend on the right. (**C – E**) The frequency of individual leg movements increases with temperature. Part (**C**) shows example frequency time series from 30 s of grooming at 18°C (top, blue outline) and 30°C (bottom, red outline). Gray dashed line = mean of this sample; green dashed line = reference at 6 Hz. (**D**) Histogram of leg movement frequencies, sampled from seven temperatures (18–30°C.) Lower temperatures are indicated in blue and higher ones in red. Thin lines—individual histograms; thick lines—average of samples at each temperature. All histograms are computed from the 84 ethograms of dust-stimulated flies in (**B**). (**E**) Box plots of leg sweep frequencies. Dots show individual fly averages, while the blue bars show the mean frequency, the red bars mark median and the green shaded areas indicate standard deviation (SD) and error (Blue/red bars in box plots – mean/median; shaded areas – SD and SE). (**F–H**) The frequency of the long time scale (ff-cycles + hh-cycles) also increases with temperature. (**F**) As described in *Figure 1D*, this plot shows the confidence in samples recorded at 18°C (top) and 30°C (bottom). (**G , H**) Similar to the panels (**D , E**) but showing the increase of ff-cycle frequency with temperature computed from the 84 dust-stimulated flies.

The online version of this article includes the following figure supplement(s) for figure 2:

**Figure supplement 1.** Frequencies of leg rubs and head sweeps increase with temperature.

**Figure supplement 2.** The hh-cycles also contract with increasing temperature.

**Figure supplement 3.** Bout alternations remain periodic across a range of temperatures.

**Figure supplement 4.** Autocorrelation analysis show temperature driven contraction across time scales.

asked if behaviors at the two time scales contract with temperature by the same amount, which could suggest a linkage between them.

## Two time scales contract together with temperature elevation

Several metrics indicate that the short and long time scale oscillations contract at the same rate as temperature increases. We noticed that the number of leg movements within an ff/hh-cycle is fairly consistent, averaging ~15: means range from 13.2 to 15.8 and s.d. range from 0.75 to 1.95 across

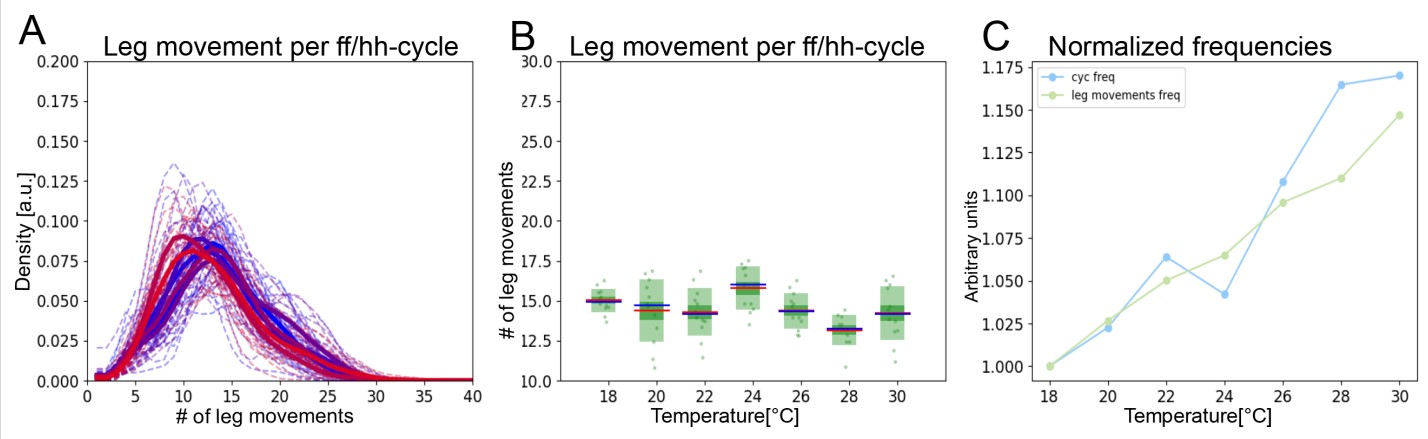

**Figure 3.** The two time scales contract together with increasing temperature. (**A**) Histograms of numbers of leg movements per ff/hh-cycle in cooler temperatures (blue) and warmer temperatures (red). (**B**) Box plots of leg movement counts per ff/hh-cycle across the seven temperatures; statistics as in **Figure 2E**. (**C**) The frequency of individual leg movements and bout alternations (ff/hh-cycles) increases roughly linearly with temperature but over different time scales (ms vs. s; 7 Hz vs. 0.5 Hz). To see if they increase at the same rate, we compare them in arbitrary units. Frequencies were normalized by dividing each mean value from **Figure 2E** by the lowest value recorded: this produces the rate of change, where 1.0 means no change and values above 1.0 reflect the increased rate. See **Figure 3—figure supplement 1** for similar effects in hh-cycles.

The online version of this article includes the following figure supplement(s) for figure 3:

**Figure supplement 1.** Head sweeps and hh-cycles contract at the same rate as temperature increases.

temperatures (**Figure 3A**). At 18°C, there are 15 leg movements of 175 ms for an ff/hh-cycle duration of 2.38 s, while at 30°C there are 15 leg movements of 152 ms for an ff/hh-cycle duration of 2.04 s. Thus, the average number of leg movements per ff/hh-cycle remains constant even as the cycle duration shortens with increasing temperature (**Figure 3B**).

An alternative way to determine whether the two time scales contract together with temperature is to plot their contraction *rates*. Normalizing by their minimal frequencies, we can visualize the slope of temperature dependence for each time scale and the correlation between them is striking ($R^2$=0.96, p<0.001; **Figure 3C**). Both time scales contract at the same rate, with $R^2$ values of 0.995 and 0.956, respectively. This analysis combined both head sweeps and leg rubs for the short time scale and ff/hh-cycles for the longer one, but when only head sweeps and hh-cycles are considered, a similar correlated contraction with temperature is also observed (**Figure 3—figure supplement 1**).

It is possible that temperature will affect both time scales of grooming behaviors at the same rate just because increased temperature tends to speed up all behaviors through its effect on neural activity, making the apparent coupling between grooming CPGs an epiphenomenon. We consider this unlikely because the two time scales can be decoupled in spontaneously grooming flies, where they respond differently to temperature, as described below.

## Periodicity and correlation between time scales persist when sensory stimulation is constant

So far, we have shown that two hallmarks of CPGs—periodicity and temperature-dependent frequency increase—hold for both short time scale leg sweeps or rubs and long time scale alternations between bouts of leg rubbing and head cleaning. An additional criterion for determining if a behavior is controlled by a CPG is that rhythmic output does not require rhythmic input. It is challenging to isolate the contribution of sensory input or feedback to the rhythms we observe in grooming. There are mechanosensory bristle neurons that detect dust and induce grooming, and proprioceptive sensors that detect limb position or movement during grooming. If these sensory inputs are rhythmic, they could contribute to both short and long time scale rhythms.

When flies are covered in dust, their own grooming actions alter sensory input stimuli. The faster their legs sweep, the more quickly dust is removed. Perhaps the coupling between time scales can be explained because faster leg sweeps result in more dust removal, which reduces sensory drive and thus shortens grooming bouts. In other words, the rhythmic behavioral output could result in

similarly rhythmic sensory input—thus not excluding a reflex chain explanation of the observed periodicity (*Harris-Warrick and Ramirez, 2017*). What happens to the periodicity of the long time scale grooming rhythms, across temperatures, if the sensory drive is held constant? We predict that if it is indeed CPG-controlled, it should still contract with increasing temperature. We test this using optogenetic activation of all mechanosensory bristle neurons.

We previously demonstrated that this manipulation induces grooming, beginning with the anterior body parts, and causing alternation between bouts of head cleaning and front leg rubbing (*Hampel et al., 2017*; *Zhang et al., 2020*). Here, we combine optogenetic activation for constant sensory input with changing temperature to show that both individual leg movements and bout-level alternations increase in frequency with temperature and that they do so in a correlated manner. The expression pattern used to activate mechanosensory bristle neurons is shown in *Figure 4—figure supplement 1*. Representative ethograms at 18°C and 30°C show characteristic alternation between bouts of head cleaning and front leg rubbing (*Figure 4A*). The entire behavioral data set of optogenetically induced grooming over a range of temperatures is shown in *Figure 4B*.

Uniform sensory input still evokes rhythmic output at both short and long time scales (*Figure 4—figure supplement 2*). For example, the PIs at 18°C for short and long time scales are 0.29 (s.d.=0.03) and 0.35 (s.d.=0.03), respectively (*Figure 1—figure supplement 3E*), and the amount of periodic behavior, considering the long time scale alterations, is 62% (s.d.=10%) (*Figure 1—figure supplement 3H*). This amount of periodic behavior is less than the 80% (s.d.=8%) in the dusted flies, and the long time scale rhythm also appears more ragged. We propose that sensory feedback may be needed to stabilize rhythms but not necessarily to generate them in the first place. The period of the optogenetically induced rhythms gets shorter with temperature (*Figure 4C–H*). The frequencies are similar to dust-induced grooming, and the average number of leg movements per ff/hh-cycle is also preserved across the range of temperatures (*Figure 4I–J*). As with dusted flies, the rate of contraction of the two time scales is correlated (*Figure 4K*), supporting the hypothesis that the short time scale leg movements and the long time scale bout alternations are both controlled by CPGs, and that these circuits are yoked together, even under constant sensory stimulation.

Optogenetic activation of mechanosensory neurons does not precisely mimic the physical stimulus of dust itself, and the response of the optogenetically stimulated flies to temperature reflects this. Both short and long time scale behaviors occur with somewhat shorter periods at lower temperature than their dust-evoked counterparts, and they stop increasing beyond 26–28°C (compare *Figures 2E, H ,, 4E and H*). One possible explanation is that the optogenetic stimulation is 'maxing out' the sensory input: it may be driving the fastest leg movements biomechanically possible, or the upper bound of the CPGs' frequency range may be reached at a lower temperature. We investigated this possibility by activating the mechanosensory bristle neurons at a constant temperature but with a range of light intensities: the frequencies of movements induced remain constant (*Figure 4—figure supplement 3*). Even starting from this higher frequency baseline, the two time scales still increase with temperature at similar rates (*Figure 4K*), indicating that optogenetic activation does not immediately induce maximum movement speeds.

## Periodicity of both time scales is preserved under reduced sensory feedback

Optogenetic activation may mask acute changes in sensory feedback as the legs contact the body or each other during grooming movements, but removing or silencing sensory neurons is a more direct test of this potential contribution to rhythmicity. To determine if leg rubs and head sweeps remain rhythmic when sensory feedback is reduced, we amputated one front leg between the femur and tibia, similar to *Berendes et al., 2016*. This eliminates distal proprioceptive feedback from the amputated leg, as well as the usual mechanosensation provided by contact between the legs or the leg and the head during rubs and sweeps. We then employed the DLC software to track the position of the stump and of the intact front leg (*Figure 5—figure supplement 1*). We found that movements of both the intact leg and the stump remained periodic, with frequencies and PIs similar to those of intact legs in dusted flies. This result further supports that at least the short time scale leg movements are indeed controlled by CPGs.

We also attempted to reduce sensory feedback using genetic reagents. We blocked chemical synaptic transmission in leg mechanosensory neurons by expressing tetanus toxin as described in

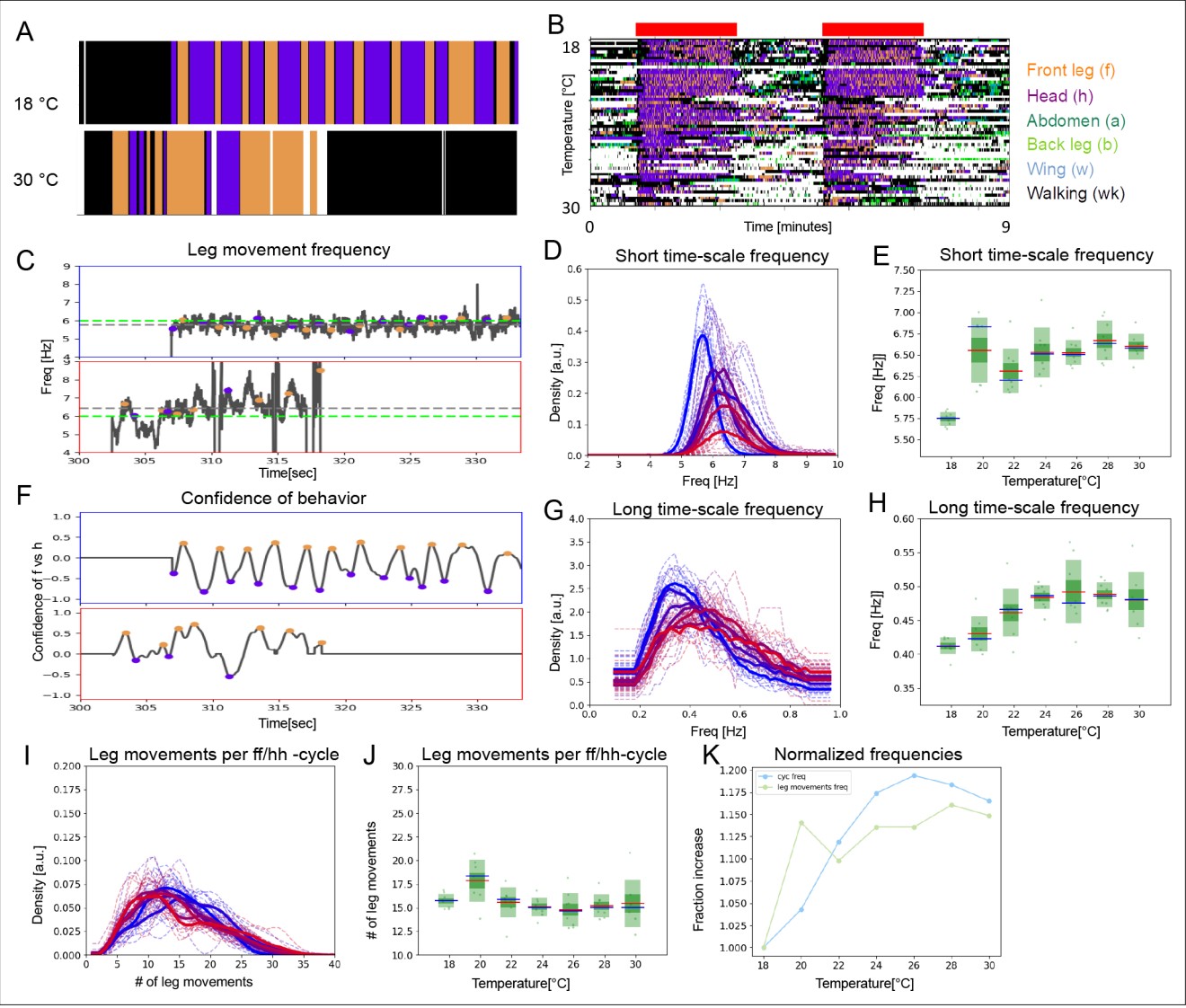

**Figure 4.** Under constant sensory stimulation, the temperature effect on both time scales persists. Undusted flies expressing the optogenetic activator *UAS-Chrimson* in mechanosensory bristles were stimulated with red light to induce anterior grooming behavior. Examples ethograms recorded at 18°C (top) and 30 °C (bottom) are shown in (**A**), while (**B**) shows the whole data set of ethograms representing 56 flies across the range of temperatures, similar to *Figure 2B*. The green bars represent the periods of light activation, lasting 2 min each, to optogenetically induce grooming. (**C**) Examples of leg movement frequencies at 18°C (top) and 30°C (bottom), (**D**) histograms of mean frequencies at cool (blue) and warm (red) temperatures, and (**E**) box plots of the increase in leg movement frequency with temperature; plots and statistics as described in *Figure 2C, D and E*; compare to short time scale effects where grooming is induced by dust. (**F–H**) Long time scale ff-cycle + hh-cycle analysis same as in *Figure 2F, G and H*. (**I**) Histograms and box plots (**J**) of median leg movement counts per ff-cycle and hh-cycle across the seven temperatures, quantified as in *Figure 3A and B*. (**K**) The rate of temperature-driven increase in frequency is shown by normalization as in *Figure 3C*.

The online version of this article includes the following figure supplement(s) for figure 4:

**Figure supplement 1.** Expression pattern of the mechanosensory bristles driver line used in optogenetics experiments.

**Figure supplement 2.** Bout alternations remain periodic across a range of temperatures in optogenetically stimulated flies.

**Figure supplement 3.** Optogenetically stimulated flies at different light intensities.

*Mendes et al., 2013* observed grooming behavior in response to dust. *Figure 5A* shows ethograms of grooming behaviors for both the experimental (TNT) and control (GFP) groups; *Figure 5B–C* shows the neurons that have been genetically inhibited. The frequency of rubs is lower in the TNT group than in the control (p< 0.001) (*Figure 5D*) while the frequency of sweeps is similar for both (*Figure 5E*). The long time scale oscillations are significantly slower in the TNT group (p<0.003) (*Figure 5H*, also

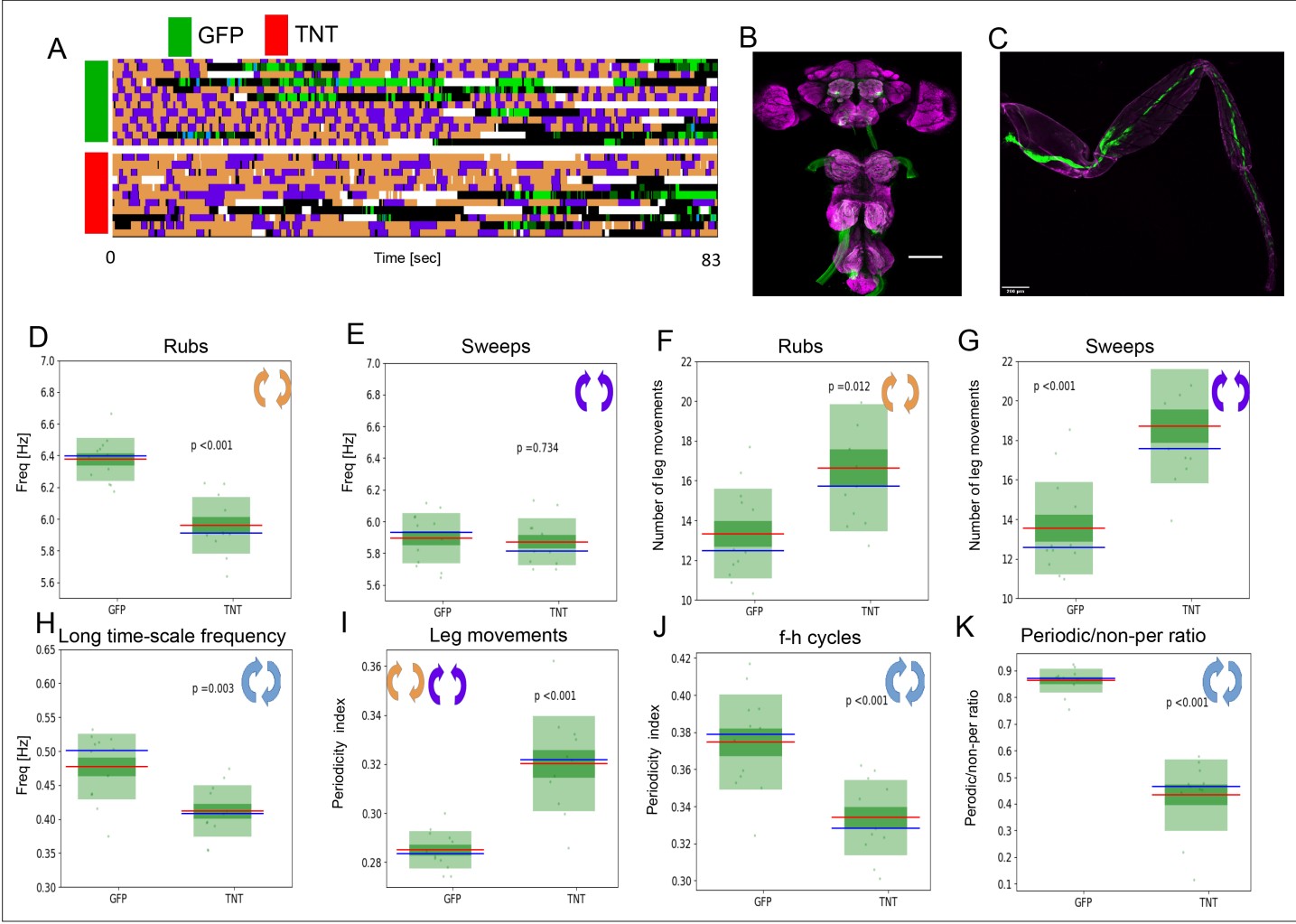

**Figure 5.** Periodicity of both time scales is preserved under reduced sensory feedback. (**A**) Ethograms of grooming behavior flies with inhibited proprioception (TNT, red bar) versus the ethograms of the control group (GFP, green bar). (**B**) Expression pattern of the leg mechanosensory neurons driver line used in TNT inhibition experiments. Expression pattern of *540-GAL4, DacRE-flp > 10X-stop-mCD8-GFP* in central nervous system. Magenta: anti-Bruchpilot. Green: anti-GFP. Scale bars, 100 μm. (**C**) Expression pattern of *540-GAL4, DacRE-flp > 10X-stop-mCD8-GFP* in leg sensory neurons. Magenta: cuticle autofluorescence. Green: innate GFP fluorescence. Scale bars, 200 μm. (**D , E**) Box plots of leg-rub and head-cleaning frequencies, respectively. (**F , G**) Similar as in (**D , E**) but for the count of rubs and sweeps per ff-cycle. (**H**) Frequencies of the long time scale (computed from ff-cycles and hh-cycles as in *Figure 2G and H*) for both groups. (**I**) Box plots of Periodicity Indexes for the short time scale of the control (GFP) and the experimental (TNT) flies. (**J**) Similar as in (**I**) but for the long time scale. (**K**) Ratios of periodic versus non-periodic long time scale oscillations in control and experimental groups. See also *Figure 5—figure supplement 1* for removal of sensory cues by amputation.

The online version of this article includes the following figure supplement(s) for figure 5:

**Figure supplement 1.** Periodicity and frequency are preserved in amputated front leg.

ethograms in *Figure 5A*) and so are the numbers of leg movements per ff/hh-cycle (*Figure 5F–G*). Both groups exhibited periodic behaviors on both the long and short time scales (*Figure 5I and J*). The sensory inhibited flies also performed less overall periodic behavior on the long time scale (*Figure 5K*). Taken together, these results suggest that grooming behaviors do not require sensory input for periodicity, but do utilize it for timing, modulation, and perhaps stabilization of the motor output, especially on the longer time scale.

## Nested CPGs can be decoupled in spontaneously grooming flies

Flies groom robustly in response to dust or optogenetically controlled mechanosensory stimulation, but they also groom spontaneously. The leg movements they perform are recognizable sweeps and rubs, and they occasionally produce alternating bouts of head cleaning and front leg rubbing as

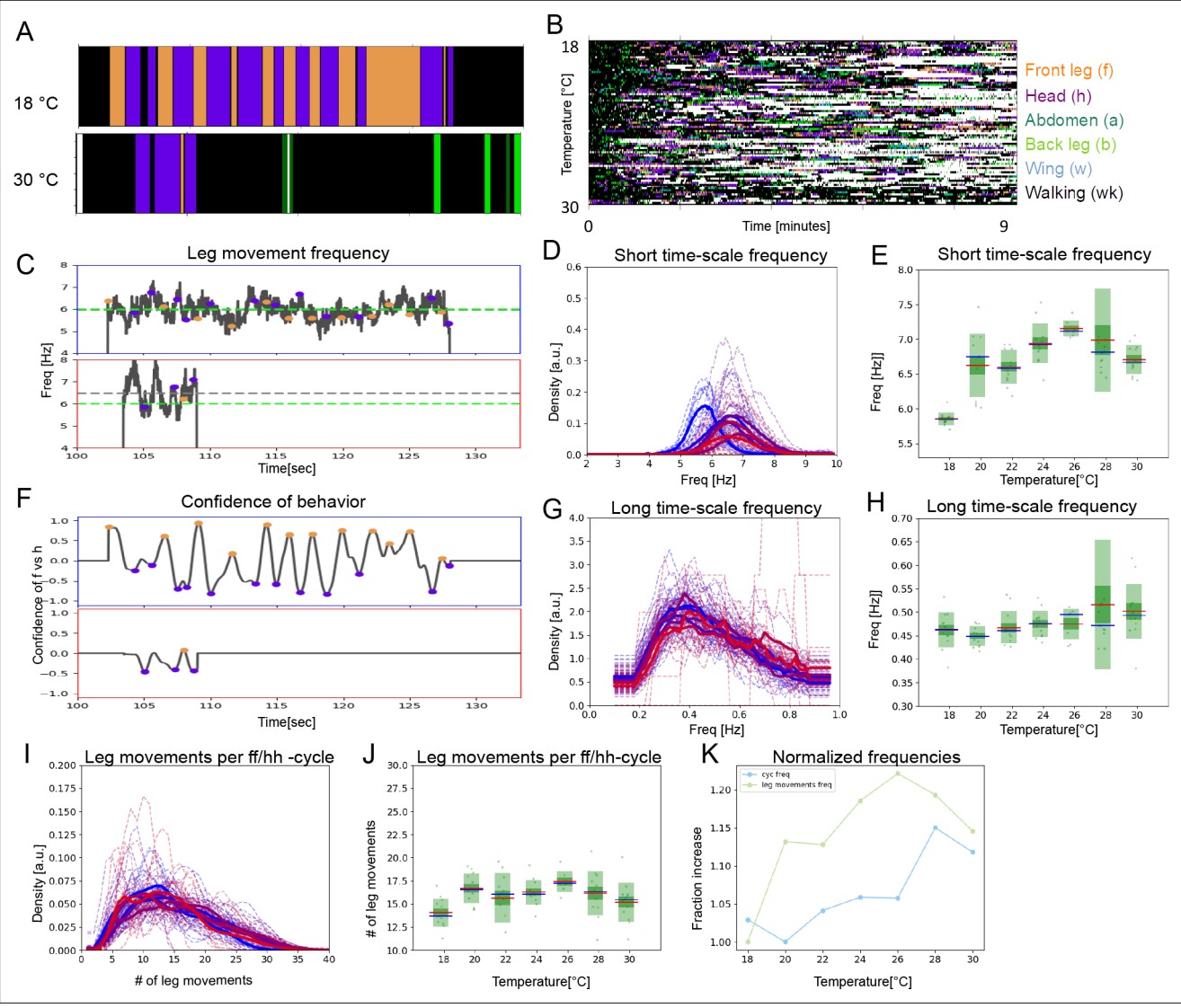

**Figure 6.** Two time scales of patterned movements can be decoupled in spontaneous grooming. Spontaneous grooming was recorded in undusted flies at a range of temperatures between 18°C and 30 °C. Examples are shown in (**A**) and the whole data set of 80 flies recorded for 13 min is shown in (**B**). (**C**) Examples of spontaneous leg movement frequencies at 18°C (top) and 30°C (bottom), (**D**) histograms of mean frequencies at cool (blue) and warm (red) temperatures, and (**E**) box plots of the increase in leg movement frequency with temperature; plots and statistics as described in *Figures 2, 3C and D*, and (**E**); compare to short time scale effects where grooming is induced by dust. (**F–H**) Long time scale (ff-cycle + hh-cycle) analysis comparable to *Figure 2*, *Figure 3F, G, and H*. (**I**) Histograms and box plots (**J**) of median movement counts per ff-cycle and hh-cycle across the seven temperatures, quantified as in *Figure 3A and B*. (**K**) The rate of temperature-driven increase in frequency is shown by normalization as in *Figure 3C*. See also *Figure 6—figure supplements 1 and 2*, and 3 for evidence of periodicity and response to temperature.

The online version of this article includes the following figure supplement(s) for figure 6:

**Figure supplement 1.** Separating periodic alternations from the non-periodic does not improve scaling with temperature in unstimulated flies.

**Figure supplement 2.** Periodicity of the long time scale across different groups of flies.

well (*Figure 6A and B*). These flies have no experimentally applied sensory stimuli—only what they generate themselves by contact between their legs and bodies, and the associated proprioceptive feedback—so these motor patterns are most likely to be generated by internal circuits. We analyzed the temperature response of both time scales of grooming movements in these spontaneously grooming flies.

Although these flies groom less than dusted or optogenetically activated flies, they show characteristic sweep and rub frequencies that increase with temperature, albeit with higher variance,

ranging from 5.85 Hz to 7.15 Hz. (*Figure 6C–E*). The ff/hh-cycles are rarer, and when they occur, the temperature-dependent contraction of the long time scale is less pronounced than in the stimulated flies ($R^2$=0.86; p=0.012) (*Figure 6F–H*). Perhaps most strikingly, although the number of leg movements per ff/hh-cycle is similar to stimulated flies (~17; with higher variation across the temperature range, *Figure 6I and J*), the correlated temperature-dependent contraction of the short and long time scales observed in the dusted and optogenetically activated flies is *not* seen in the spontaneously grooming ones ($R^2$=0.43; p=0.332) (compare *Figure 6K* to *Figures 3K and 4K*). The frequency of leg rubs and sweeps increases with temperature at a greater rate than the frequency of ff/hh-cycles (*Figure 6K*), suggesting that the pattern-generating circuits that control the two time scales of movements may be dissociated in spontaneous grooming.

Perhaps, in spontaneously grooming flies, the two levels of nested CPG are linked only sporadically, when the flies engage in more periodic behaviors. To explore this possibility, we separated the periodic long time scale oscillations from the non-periodic ones using a threshold of ACF shoulder peak prominence (see Materials and methods and *Figure 1—figure supplement 2*). We hypothesized that if the periodic behaviors are CPG-driven and the non-periodic behaviors are not, then the periodic behaviors should scale with temperature more strongly than non-periodic ones. However, we found that neither periodic nor non-periodic behaviors changed frequency with temperature (*Figure 6—figure supplement 1C* and D) and there was no significant correlation between the two time scales (*Figure 6—figure supplement 1E* and F), or between the two levels of the nested CPG, in spontaneously grooming flies.

## Discussion

Temperature manipulations have been instrumental in identifying behaviors controlled by CPGs and for locating where those circuits reside. Robust rhythms are retained across a range of temperatures in the crustacean stomatogastric ganglia, where the relative timing of events is preserved even as the sequence itself changes speeds (*Alonso and Marder, 2020*; *Rinberg et al., 2013*; *Tang et al., 2012*). Local cooling of the cat spinal cord demonstrated which segments contain locomotion control circuits (*Deliagina et al., 1983*), and local warming showed that cricket chirping is governed by thoracic rather than brain ganglia (*Pires and Hoy, 1992a*; *Pires and Hoy, 1992b*). The role of CPGs in bird song, and the importance of the HVC brain region for sequence timing, was shown because local cooling expands the entire song without changing the relative durations of the syllables and sub-syllabic components (*Armstrong and Abarbanel, 2016*; *Long and Fee, 2008*). In frogs (*Xenopus laevis*), both local cooling and ambient temperature affect the frequencies of vocalization patterns (*Yamaguchi et al., 2008*). Here, we follow this tradition to examine fly grooming behavior at different temperatures, demonstrating that both its short and long time scale components show evidence of CPG control. Although here we change the temperature of the whole fly, our study opens the way to use anatomical and genetic tools to map underlying neural circuits in future experiments.

Fly grooming is an innate motor sequence with both repetition and flexibility. Behavior analysis has shown organization over several time scales, from single stereotyped leg movements and alternations between bouts of repeated actions targeting specific body parts (*Mueller et al., 2019*), to a gradual and probabilistic progression from anterior toward posterior grooming (*Seeds et al., 2014*). Here, we investigate what aspects of fly grooming are periodic, demonstrating that the short time scale leg sweeps and rubs repeat at characteristic frequencies, in the absence of patterned sensory input, and in a temperature-dependent manner (*Figure 2D–E*), consistent with control by CPGs. Since the neural circuits that constitute the CPGs controlling these periodic leg movements are likely to overlap with those proposed to coordinate walking (*Bidaye et al., 2018*), this was not unexpected.

But our more surprising discovery was that the alternation between bouts of head cleaning and front leg rubbing is also periodic (*Figure 1—figure supplement 4A*). While this oscillation is more variable, it too, is independent of patterned sensory input, and increases in frequency with temperature (*Figures 2G–H and 4G–H*)—suggesting that there may be an additional CPG operating at this longer time scale. This high-level, overarching, CPG can control the alternations between leg rubs and sweeps that are themselves governed by faster, low-level CPGs. The idea that multiple CPGs coordinate movements is not new: CPGs may control each leg joint, regulating the interaction between flexor and extensor muscles, governing the way coxa-trochanter and femur-tibia joints are coordinated to produce forward or backward walking, or mediating interactions among limbs (*Feng et al., 2020*;

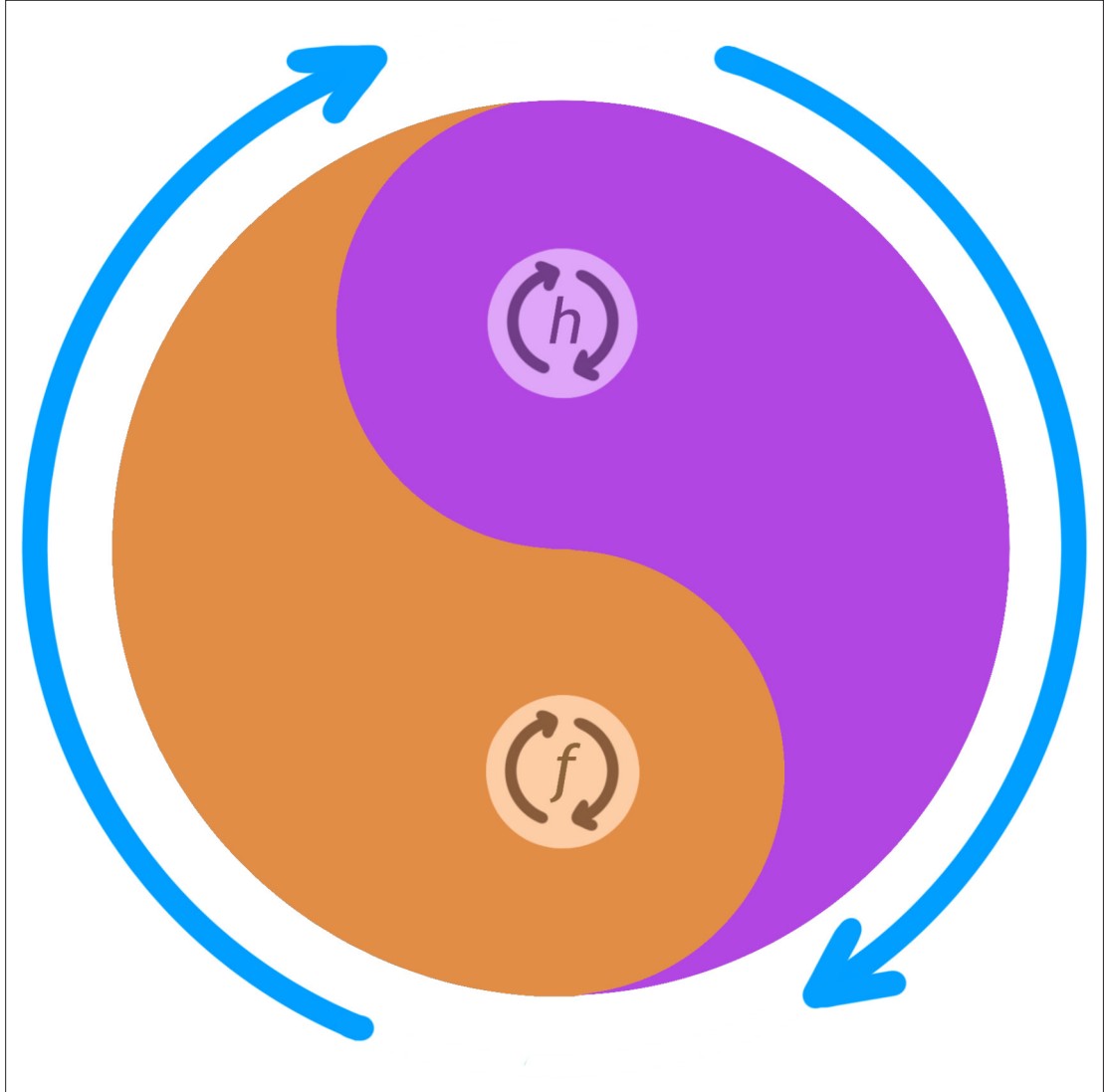

**Figure 7.** Conceptual schematic of nested CPG controlling anterior grooming behavior. This diagram illustrates how the lower level CPG modes controlling individual leg rubbing (f) and head cleaning (h) oscillations (small orange and purple arrows) can be coordinated by an over-arching, high-level, CPG controlling the alternation between the front leg rubbing and head cleaning bouts (large blue arrows). The probability of f versus h is indicated by the thickness of their corresponding colors (orange vs. purple) at each angle of the circle. For example, at 5 o'clock the probability of f is higher than that of h and at 11 o'clock probability of h is higher than that of f, and so on. CPG, central pattern generator.

*Mantziaris et al., 2020*). In the vertebrate spinal cord, inhibitory and excitatory commissural neurons can cause the CPGs controlling the legs to synchronize for a hopping gait or operate out-of-phase for walking (*Kiehn, 2016*). CPGs in frogs have been shown to act on different time scales (sequentially) to govern their vocal behaviors (*Zornik et al., 2010*), and leeches can alternate between different CPG-driven behaviors (*Esch et al., 2002*). The concept of nested CPGs has recently been extended to explain the flexible coordination of behaviors ranging from fish swimming to bird song (*Berkowitz, 2019*). We map fly grooming behavior into this nested CPG framework as shown in *Figure 7*.

In a recent publication, nested CPGs similar to the one we are proposing here have been described in the crab stomatogastric ganglion, where fast pyloric and slow gastric mill rhythms are coupled and compensate for temperature changes (*Powell et al., 2020*). The combination of two rhythmic behaviors raises the possibility that the governing circuits interact, like meshed gears of different sizes, to simplify control of a repetitive behavior on multiple time scales. Just as the relative number of rotations between different gears is the same regardless of the absolute speed of the mechanism, so the phase of different components of CPGs is conserved across a range of temperatures (*Harris-Warrick*

*and Ramirez, 2017*). In the case of fly grooming, this relationship between fast and slow components of the system may be reflected in the constant number of leg movements within each grooming bout across a range of temperatures (*Figure 3*). The fast and slow components of the nested CPG can be coupled (in stimulated flies) or decoupled (in spontaneously grooming flies). It is also possible that coupling can occur dynamically over the period of behavior execution, resulting in alternation between periodic and non-periodic behaviors on the long time scale (*Figure 6—figure supplement 2B*). However, in spontaneously grooming flies, we saw no coupling between the two time scales in either periodic or non-periodic behavioral episodes (*Figure 6—figure supplement 1*). The neural circuit implementation of nested CPGs, and the potential 'clutch' mechanisms that engage and disengage their connections, remain to be determined.

Dust-induced grooming can be routine, rhythmic, repetitive, and governed by CPGs, but spontaneous grooming may have more varied control architecture. Fast walking insects may be more likely to engage CPGs, while slower ones may rely more on sensory feedback and reflex chains (*Mantziaris et al., 2020*). Just as you can walk a straight path without thinking about it or you can carefully place each foot on icy terrain, flies may have alternative ways to produce movement sequences. Disturbing a single bristle elicits a single directed leg sweep (*Kays et al., 2014*; *Vandervorst and Ghysen, 1980*), while covering the fly in dust evokes an entire grooming program with bouts of several leg sweeps and rubs, alternation between the bouts, and slow gradual anterior-to-posterior progression (*Phillis et al., 1993*; *Seeds et al., 2014*). Our observation that spontaneous grooming shows less periodicity at the long time scale (*Figure 6—figure supplement 2B*) can be interpreted as effective decoupling of the two levels of the nested CPG in those flies when they engage in more sporadic and shorter episodes of grooming.

In both optogenetically stimulated flies and in flies where sensory feedback was inhibited, we observe less periodicity of the long time scale than in the dust-stimulated flies (*Figure 1—figure supplement 2H* and *Figure 5K*). We speculate that sensory feedback is needed to stabilize and modulate the rhythmic behavior, even though it is not necessary to produce it. For example, a fly could produce in-phase rhythms to provide the general control of head-cleaning movements but the sensory feedback from various parts of the head could adjust these movements to optimize the efficiency of cleaning of those parts. The concept of sensory feedback playing a role in stabilizing and adjusting rhythmic behaviors has been explored in walking mice (*Mayer and Akay, 2018*, *Santuz et al., 2019*) and flies (*Mendes et al., 2013*), as well as in robotics (*Wang et al., 2014*, *Sartoretti et al., 2018*) and computational modeling of CPGs (*Yu and Thomas, 2021*).

Nested CPGs can be implemented to generate arbitrarily complex behaviors that are also periodic and well-coordinated. Many different time scales of simple periodic movements can be combined into a more complex periodic behaviors the way different frequencies of sine waves can be combined to create arbitrarily complex shapes in Inverse Fourier Transformation. If CPGs can operate across various time scales in this fashion, then all that is needed to create arbitrarily complex yet highly reproducible behavior is to adjust the weight of each contributor to a multi-level nested CPG.

The specific neural circuits that constitute CPGs have been challenging to identify in most preparations and even the best characterized would benefit from more comparators. The electrophysiological recordings that have been so critical in CPG analysis in other preparations are possible but challenging in *Drosophila*, but the real strength of the system is the ability to demonstrate that specific neurons have a causal connection to rhythm generation. New anatomical resources to map neural circuits, especially the complete electron microscopy data set covering the ventral nerve cord (*Phelps et al., 2021*), will enable the identification of the pre-motor neurons most likely to participate in CPGs and the commissural connections that may mediate among them. Functional imaging of neural activity in dissected, fictive preparations (*Pulver et al., 2015*) and even in behaving flies (*Chen et al., 2018*) is another promising approach to identify neurons with rhythmic activity that may constitute parts of the CPGs. The genetically encoded calcium indicators and new voltage sensors have fast enough onset and offset kinetics to capture rhythmicity on the expected time scales (*Simpson and Looger, 2018*). For example, whole nervous system monitoring of neural activity was recently described in *C. elegans* to identify hierarchical and dynamic control of nested locomotion patterns (*Kaplan et al., 2020*).

Rhythmic activity and circuit connectivity may suggest candidate neurons, while genetic tools to target specific neurons (*Jenett et al., 2012*) and optogenetic methods to impose altered activity patterns (*Klapoetke et al., 2014*) present a way to causally connect them to control of rhythmic

behaviors. The behavioral evidence presented here suggests that a two-level nested CPG can control aspects of fly grooming over at least two time scales. Identifying what circuits constitute these CPGs and mapping the neurons that coordinate their interactions is feasible, now that we know we ought to be looking for them.

## Materials and methods

### Fly stocks and husbandry

*Drosophila melanogaster* was reared on common cornmeal food in a 25°C incubator on a 12-hr light/dark cycle. Three to five days, *CantonS* males were used for dusting and spontaneous grooming experiments. For optogenetic experiments, larvae were raised on normal food. After eclosion, 1-day-old adults were transferred into food containing 0.4 mM all-trans-retinal and reared in the dark for another 2 days.

*R74C07-GAL4* (Bloomington Stock Center 39847), *20XUAS-IVS-CsChrimson.mVenus* (Bloomington Stock Center 55134), *5–40* GAL4 (*Hughes and Thomas, 2007*), *DacRE-FLP* (*Mendes et al., 2013*), *UAS>stop>* TNT (*Stockinger et al., 2005*), and *10XUAS>stop>myr-GFP* (Bloomington Stock Center 55811) were used in the paper. Ten-day-old males were used in TNT inhibition experiments. Three- to five-day-old males were used in other experiments.

### Behavior experiments with temperature control

Behavior videos were recorded inside New Brunswick I2400 incubator shaker or DigiTherm DT2-MP-47L heating/cooling incubator. Experiments were performed every 2°C between 18°C and 30°C. Temperature was monitored by a Govee H5072 Bluetooth thermometer. For dusting experiments, the room temperature was also adjusted to the target temperature to make sure flies stay at the same temperature during fly dusting. Each fly was tested once in one condition. Three types of chambers were used in fly dusting assay: dusting chamber (24-well corning tissue culture plate #3524), transfer chamber, and recording chamber. Flies were anesthetized on ice and transferred to the middle four wells of the transfer chamber. Transfer chamber with flies was then kept in the incubator set to the target temperature for 15 min. For fly dusting, around 5 mg Reactive Yellow 86 dust was added into each of the four middle wells of the dusting chamber. Transfer chamber was aligned with the dusting chamber. Flies were tapped into the dusting chamber and shaken 10 times. After dusting, flies and dust were transferred back into the transfer chamber. Transfer chamber was banged against an empty pipette tip box to remove extra dust. Dusted flies were then immediately tapped into the recording chamber for video recording. The whole dusting process was performed in a Misonix WS-6 downflow hood.

For optogenetics and spontaneous grooming experiments, ice-anesthetized flies were put into the recording chamber directly. Recording chamber with flies was then kept in the incubator set to the target temperature for 15 min before the experiment. In spontaneous grooming experiments, flies were shaken five times after the 15 min incubation. They were then rested in the incubator for another 3 min before recording.

60 Hz videos were recorded for 9 min in optogenetics experiments, 13 min in dusting, and spontaneous grooming experiments with a FLIR Blackfly S USB3 camera. Each video is recorded from the top and it captures flies in four chambers. Infrared backlight was used for all experiments. Custom-made LED panels (LXM2-PD01-0050, 625 nm) were used for optogenetic activation from below. 20 Hz 20% light duty cycle was used in optogenetics experiments. Red light intensity was adjusted to 0.85 mW/cm$^2$. Before the video analysis, the frames were split into four quadrants corresponding to the four chambers.

For all experiments described above, 8–12 flies were used per temperature or per experimental group (for the TNT inhibition experiment). Occasionally a recording chamber was empty or the fly was visibly damaged or dead. Those chambers were excluded from further analysis. These numbers were sufficient to extract the quantities we use in this work such as ff/hh-cycle periods.

### High-resolution videos recording

For high-resolution videos used for leg tracking, flies were put in a 10-mm diameter quartz chamber, and 100 Hz videos were recorded from below. An FLDR-i132LA3 red ring light (626 nm) was used for

optogenetics activation. For leg amputation experiments, one front leg was amputated at the middle of the femur. Flies were recovered for 3 days before the experiments. The dusting procedure is the same as what is described above.

## Leg movements counting from video

We did not track individual legs or joints so the frequencies of leg movements during grooming were estimated from frame-to-frame changes in pixel values. When flies are walking or engaged in other non-grooming behaviors, such measurements would not be very useful. However, during grooming, flies are standing in one place and the most intense pixel value differences roughly correspond to those created by leg movements (rather than whole body translations).

Regions of interest were cropped around the animals to produce 80×80 arrays of pixels (**F**). The **F** arrays were reshaped into 1600 columns and 32 rows, corresponding to 32 consecutive frames (~0.5 s window at 60 Hz), and were stacked together to obtain 32×1600 array of pixel values (**W**).

The derivatives of **W** across time were computed using Numpy.gradient() function to obtain matrix of frame-to-frame differences **D**. The matrix **D** was smoothed using Gaussian filter (scipy.ndimage.filters.gaussian_filter). Sigma=[2, 5]. This was done to denoise the signal—to remove non-biological high-frequency changes (temporal smoothing) and to reduce spatial resolution (so that more neighboring pixels capture the same signal). All the values of **D** where cumulative change of pixel values was less than threshold ($\theta$=5.0) were converted to zeros to retain only those pixels where significant movement occurred. An example of **D** matrix is shown in *Figure 1—figure supplement 1*.

Notice in the figure the several non-zero columns (corresponding to pixels) that appear sinusoidal. These columns contain the signals that have been smoothed to remove the high-frequency noise and are prominent enough to 'survive' the thresholding. Therefore, we assume that the movements represented in these columns correspond to relatively large, periodic actions that occur while the animal is standing (otherwise the sinusoidal signals would not stay in one column over the time window). Also, only the frames where grooming behavior was detected by the ABRS were considered for leg movement counting. This assumption leads us to conclude that the periodic signals are caused by periodic leg movements during grooming. So the counting the peaks across the columns of matrix **D** would then result in the number of times a leg crossed a particular pixel. The peaks were found applying the scipy.signal.find_peaks() function across the rows on the matrix **D**. The peaks were then counted for each column in D and the median of all the counts was finally taken to be the number of leg sweeps or rubs that occurred in that time window (0.5 s). Note that at the frame rate of 60 Hz, we should be able to detect frequencies of periodic signals below 30 Hz and the highest frequency of leg movements we observed was below 7 Hz.

The number of leg movements per second was used as the leg movement frequency.

## Automatic behavior recognition from videos

Probabilities and ethograms of grooming behaviors were extracted from raw videos using ABRS. For detailed description see *Ravbar et al., 2019* and for the most updated version see ABRS (https://github.com/AutomaticBehaviorRecognitionSystem/ABRS; *Ravbar, 2021*; copy archived at swh:1:rev:7c558561ae82b62bfcc5337a334b07efcbb18da2).

Briefly, ABRS pre-processes video data by compressing it into spatio-temporal features in the form of three-channel space-time images (3CST images, shape=80×80×3) where the first channel [0] contains raw video frame pixel values [80×80], the second channel [1] contains spectral features of pixel value dynamics over a time window of 16 frames, and the third channel [2] contains frame-to-frame difference (frame subtraction). The 3CST images are classified into seven behavioral categories (front leg rubbing, head cleaning, back leg rubbing, abdominal cleaning, wing cleaning, and whole-body movements [walking]) by ConvNet (Covolutional Neural Network – CNN) implemented in Tensor Flow (in Python) (https://www.tensorflow.org), using a model trained with diverse set of videos of fly grooming behaviors. In the final output layer of the CNN are the probabilities of the grooming behaviors. In this work, we focus entirely on the probabilities of front leg rubbing and head cleaning behaviors: P(f) and P(h).

## Behavioral confidence

The long time scale oscillations are quantified as probabilities of behaviors obtained from the ABRS CNN described above. Behavioral confidence (BC) is computed as:

$$BC = P(f) - P(h) \tag{1}$$

The BC signal is smoothed two consecutive times with time window of 31 frames (0.5 s) using the scipy.signal.savgol_filter function to remove high-frequency noise.

## Autocorrelation and periodicity analysis

ACFs were computed from a 0.5-s time window for individual leg movements (leg rubs and sweeps). The signal for autocorrelation was extracted from raw movies as follows: (1) Matrix **D** was obtained as described above (Leg movements counting from video); (2) ACFs were computed for every column of **D** (an example of the ACFs and the matrix **D** are shown in *Figure 1—figure supplement 1*); and (3) All the ACFs were averaged to obtain the mean ACF for that time window. The mean ACFs were stacked to obtain the ACF array with dimensions 60×F, where F is the number of frames in the movie. The AFCs were computed using scipy.signal.correlate function from SciPy library.

ACFs for the long time scale (the oscillations between bouts of leg rubbing and head cleaning) were computed from the BC time-traces described in *Equation 1*, in the time window of 999 frames (16.6 s), also using the scipy.signal.correlate function.

To quantify the periodicity of each time scale, we computed the PI from the ACFs, defined as the ratio between the height of the prominent peak nearest to zero lags and the height of the central peak (at zero lags) of an ACF (*Figure 1—figure supplement 2*). The peaks were detected by scipy.signal.find_peaks function. The threshold for 'prominence' was set at 0.2. The threshold for height was set to 0.2. We used the same method and parameter settings for both time scales to compute the PI.

We managed to separate periodic behaviors from the non-periodic by using the scipy.signal.find_peaks function. If the first neighbor of the central peak of the ACF fell below the 0.2 'prominence' threshold we classified the pertaining behavior as non-periodic and as periodic otherwise. This is shown in *Figure 1—figure supplement 2B*.

## Frequency computations for the long time scales

BC traces were used to compute the frequencies of the long time scale.

$$f = 1/L \quad \text{Eq. 2.}$$

In Eq. 2, the L is a vector containing the lengths of periods measured from the BC traces. The peaks and valleys were detected by scipy.signal.find_peaks function, where the 'prominence' was set at 0.1 and the minimum height was set at 0.05 (valleys were found from the same signal as the peaks but multiplied by [–1] to invert it). The periods were measured either between two peaks of the BC traces (ff-cycles) or between two valleys (hh-cycles), or we combined the periods from both, the ff-cycle and hh-cycle (also see *Figure 1E* for illustration). To avoid capturing very long transitions between the peaks/valleys (which occur between episodes of grooming behavior and walking or standing), we set the maximum duration of ff/hh-cycles to be 280 frames (~4.7 s). In the sensory inhibition experiment where we compare the TNT mutants to the control we increased the duration threshold to 380 frames (~6.3 s) because the ff/hh-cycles are extraordinarily long (see *Figure 5A*).

This produced a vector of frequencies of dimensions 1×F, where F is the number of frames in a movie, corresponding to one fly. The mean and median frequencies for each fly were computed as means/medians of vector f.

## High-resolution behavior analysis

To confirm our observations of grooming behavior, its periodicity and the effect of temperature, we used an independent method for limb tracking, the DLC (*Mathis et al., 2018*) on a small sample of high-resolution videos described in the previous section. The DLC allowed us to label (virtually) three body parts on each front leg and a reference point (see *Figure 5—figure supplement 1* A showing the labels on the intact front leg). We could then track these labels across time. For each leg, we computed the angle between Tibia and Femur and tracked the derivatives of these angles across

time—examples are shown in *Figure 1—figure supplement 4A*. On these time-traces, we applied the autocorrelation analysis and FFT to compute ACFs and their spectra. (For the amputated fly, we used the relative positions of the joints and the stump instead of the angles.) We computed the PI from the ACFs.

In these analyses, we did not have access to the BC (see the section above) so, in order to examine the periodicity and frequencies of the long time scale, we computed the equivalent to BC as follows: the distance between tarsal segments was used as the 'confidence of head cleaning' and the angle between Tibia and Femur was used as the 'confidence of front leg rubbing.' We obtained the behavior confidence signal by subtracting the latter from the former. We applied autocorrelation and FFT analysis this 'behavior confidence' to estimate the PI and the frequency of the long time scale. Due to cumbersome nature of these analysis and the lack of behavioral recognition, we performed them only on three different movies from the dust-stimulated flies (*Figure 1—figure supplement 4D-F*).

## Handling of outliers and missing data points

We found and did not eliminate an outlier in optogenetically stimulated flies (leg movement frequencies at 20°C). In cases where there were missing data points (no fly was in the recording chamber), those were not counted in the statistics.

## Acknowledgements

The authors thank members of the Simpson, Louis, and Kim labs for feedback, and especially Li Guo and Dr. Josh Mueller for excellent suggestions. The authors thank Carla Ladd for creative input with figure design and Aleks Labuda for instrument design for high-resolution video recording. This research was supported by NIH-R01NS110866 and HHMI Janelia transition funding.

## Additional information

### Funding

| Funder | Grant reference number | Author |
| --- | --- | --- |
| National Institute of Neurological Disorders and Stroke | R01NS110866 | Julie H Simpson |

The funders had no role in study design, data collection and interpretation, or the decision to submit the work for publication.

### Author contributions

Primoz Ravbar, Conceptualization, Data curation, Formal analysis, Investigation, Methodology, Software, Validation, Writing – original draft; Neil Zhang, Investigation; Julie H Simpson, Conceptualization, Funding acquisition, Project administration, Supervision, Writing – review and editing

### Author ORCIDs

Primoz Ravbar http://orcid.org/0000-0001-7769-8687
Neil Zhang http://orcid.org/0000-0003-3397-5974
Julie H Simpson http://orcid.org/0000-0002-6793-7100

### Decision letter and Author response

Decision letter https://doi.org/10.7554/eLife.71508.sa1
Author response https://doi.org/10.7554/eLife.71508.sa2

## Additional files

### Supplementary files

• Transparent reporting form

## Data availability

All data is included in the manuscript or uploaded to Dryad. All computer code is available on GitHub.

The following dataset was generated:

| Author(s) | Year | Dataset title | Dataset URL | Database and Identifier |
|---|---|---|---|---|
| Ravbar P, Zhang N, Simpson J | 2021 | Behavioral probabilities and sample videos for "Behavioral evidence for nested central pattern generator control of *Drosophila* grooming" | https://doi.org/10.25349/D9QW4J | Dryad Digital Repository, 10.5061/dryad/D9QW4J |

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
