## [Editor Report]

This study presents an intriguing example of natural rhythmic leg movement oscillations at a fast time scale embedded within a slower rhythmic behavioral oscillation and the similar effects of temperature on the rates of the two rhythms. The data and the writing are both exceptionally clear and advance our understanding,

---

## [Decision Letter]

**Decision letter after peer review:**

[Editors’ note: the authors submitted for reconsideration following the decision after peer review. What follows is the decision letter after the first round of review.]

Thank you for submitting your work entitled "Behavioral evidence for nested central pattern generator control of *Drosophila* grooming" for consideration by *eLife*. Your article has been reviewed by 3 peer reviewers, and the evaluation has been overseen by a Reviewing Editor and a Senior Editor. The reviewers have opted to remain anonymous.

Our decision has been reached after consultation between the reviewers. Based on these discussions and the individual reviews below, the consensus view is that additional experiments would be required to address major concerns. As these experiments would likely require more than a two month timeframe, we are turning the submission dowm in its current form. If it is possible to address the major concerns outlined below, we would consider a fresh submission, if you would like to consider *eLife*.

The reviewers agreed that the notion of nested CPGs for coordination of grooming is important but that stronger demonstration is required that the rhythms are indeed generated by a CPG in the absence of sensory feedback. In addition, better evidence that the two rhythms are coupled would significantly strengthen the study. Key experimental suggestions are below. Full comments of the reviewers are attached to provide further guidance.

1) Better quantification of rhythms:

a) High res leg tracking will help quantify both rhythms with more accuracy. It could also be used to better describe the coupling between the two rhythms (Figure 3 and 4, what are the "leg movements" that are invariant across temp).

b) Poor confidence levels of the slow rhythm in Figure 4 and 5 need to be addressed with supporting videos, complementary analysis or new data from high res tracking.

2) Role of proprioceptive feedback in rhythm generation:

Proprioceptive feedback from the legs could be the major rhythmic input that drives both rhythms and its role should be explored by neuronal silencing (e.g. Mendes et al., 2013 *eLife*).

*Reviewer #1:*

Ravbar et al. have carried out behavioral analysis of *Drosophila* grooming behavior and have proposed that nested CPG modules are involved in generating the sub-routines during anterior grooming. It is a conceptually exciting hypothesis and is supported by authors' observation of two nested behavioral rhythms, a fast leg movement rhythm (5-7Hz) and a slower behavioral state switch rhythm (0.3-0.6Hz). The authors then use temperature change as a tool to prove existence and coupling of the underlying nested CPG modules that govern these rhythms.

This work is conceptually interesting with a potential for broad interest, and I'm in favor of its publication provided the authors address the major points.

1) The authors use temperature change as a tool to prove the existence and coupling of nested CPGs. Although there is a certain simplicity and elegance in this approach, it needs some more justification. Temperature directly influences neural activity (not restricted to CPGs). The authors cite important studies where temperature was used to pinpoint CPGs, but in these studies local heating/cooling of specific parts of the nervous system was used as a tool to increase/decrease neural activity in those areas and thereby locate the position of CPGs (e.g. Long and Fee 2008)). Global heating/cooling (as conducted in the current study) was only used in StG papers (Marder lab) to characterize the effect of temperature on previously well-described CPG networks. Given the pivotal role of these temperature change experiments in supporting central results of this paper, it needs more justification, both at a descriptive level as well as more characterization. E.g. it would be interesting to see how other attributes already tracked in these experiments (walking, hind grooming etc) are affected by temperature. Is there any behavior that is not proportionally affected by temperature and what does it tell us that is special about the anterior grooming rhythms? Also, given global heating will affect all neurons similarly, proportional changes in the two rhythms need not indicate "coupled" rhythms. This caveat should be discussed while discussing results from Figure 3.

2) A more direct evidence for CPGs would be to show that rhythmic motor output could be achieved without rhythmic sensory input and the authors also claim to have addressed this using optogenetics (Figure 4). The authors suggest that the changing quantity of dust on the fly's body as it grooms, could be a rhythmic input. This could be better explained. I'm guessing what matters is the difference in sensory input coming from head (H) versus front legs (F). If H>F then the fly does behavior "h", else it does "f". Doing behavior "h" reduces sensory input "H". With this assumption, the H-F input could indeed be rhythmic, and then the opto experiment rationale seems justified. However, for the fast rhythm (leg sweeps), the major rhythmic input is likely the proprioceptive input from the moving legs and not the continuously decreasing quantity of dust on the fly's body and hence this rhythm is not really addressed by the opto experiment. In fact the proprioceptive input from the legs could also be the defining input for the slower rhythm given that "number of leg movements" remain constant across temperatures (Figure 3 and 4K). One way to investigate pure centrally driven rhythms could be to eliminate rhythmic sensory feedback by leg amputation (e,g Berendes et al., J.Exp.Biol, 2016) or neuronal silencing (Mendes et al., *eLife*, 2013) and then apply dust/opto-stim and see if both fast and slow rhythms still persist.

3) Finally, the authors describe an interesting "decoupling" of the two rhythms during spontaneous grooming. The authors observe similar leg movement frequencies, however the longer f-f and h-h rhythms seem very rare and variable. So the "decoupling" between the two rhythms is in fact driven by "absence" of the slower rhythm in the experimental conditions. This is also interesting but should be discussed appropriately. Looking at the full ethogram in Figure 5B, it is apparent that as temperature increases, flies spend more time walking versus grooming and so the grooming bouts are likely too short to show any slow rhythms. So in theory, the rhythms could be still coupled, just now more disrupted because of a competing "walking" state.

4) The leg movement analysis needs to be justified more than what it is in the Methods section. The authors acquire videos of grooming flies at 60 Hz, then select frames containing anterior grooming bouts and in these selected frames, they then use pixel difference in consecutive frames and use the frames depicting the largest movement for quantifying frequency. It is unclear what aspect of leg movement the authors are quantifying in this analysis and whether it is affected by video frame-rate or camera angle. What motor action is represented by the "leg movement" frequency? Why is this frequency value higher for "f" bouts versus "h" bouts? Would the authors get the same results if videos were captured at a higher frame rate? It is important to disambiguate this, because the authors use "number of leg movements" as an attribute in Figure 3 to prove coupling between the CPGs. The authors could acquire some additional videos, preferably at a high frame rate, and use one of the recent tools (Pereira et al., 2019, Mathis et al., 2018) to track the legs during anterior grooming, as has been done for leg movement rhythm quantification during walking. If this type of high resolution analysis also suggests a similar leg sweep frequency, the current analysis framework of the authors will be justified. Moreover, it could provide a better understanding of results in Figure 3 and 4K.

5) For the slow f-f and h-h rhythm quantification the authors use an interesting approach (P(f)-P(h)). Although this looks promising in Figures 1D, 2F and supp Figure 2, many of the peak/trough values tend to be very close to zero in Figures 4F, 5F making it unclear what rhythms does this really represent in terms of actual behavior. Are we still talking about clear rhythms between 2 states as depicted in Figure 1A? Performing oscillation analysis using confidence values between artificially defined binary states can enforce the output to look like an oscillation and hence it is important to explicitly discuss this.

*Reviewer #2:*

This manuscript uses temperature and optogenetic manipulations to alter the speed of grooming behaviors and examines the resulting changes in fast- and slow-time scale behavioral structure. The authors find that temperature increases the frequency both of individual leg movements, and the alteration between bouts leg rubbing and head grooming, such that the number of leg cycles per bout is approximately the same across a range of temperatures. A similar results is found when grooming is elicited by optogenetic stimulation of mechanosensory bristles, rather than by dust. Finally, the authors show that coordination between these two levels of behavioral organization breaks down during spontaneous grooming. The authors draw conclusions about the control of these behaviors by nested central patterns generators. The behavioral observations presented here are interesting but I felt that more analysis of genetic or circuit manipulations would be required to firmly draw the conclusions the authors would like to about the neural control of these behaviors.

A central claim of the manuscript is that *Drosophila* grooming may be controlled by nested CPGs. As currently presented the evidence is not especially strong, as it is not clear what the alternative to central pattern generator control of these behaviors would be. The manuscript would be stronger if the authors identified neurons whose activation or inactivation altered behavioral rhythms or the coupling between fast and slow-time scale structure in the behavior.

In the case of high temperature + constant sensory stimulation the behaviors seem much more variable and irregular but this is not noted or discussed.

Line 35-36: "Automating the sequence by calling it actions in series produces reliable control" This seems like a bit of straw man. Most rhythmic behaviors such as walking, breathing, etc, are assumed or known to arise from central pattern generators, and not from individual cycle-by-cycle decision, although of course each behavior can be modulated on a cycle-by-cycle basis.

Line 73: "characteristic frequency of 6Hz." It seems in 1C that the characteristic frequency of head grooming and leg rubbing are slightly different. Can those averages be shown or given here and are they significantly different?

Line 125 + figure 2C/4C. The authors note that the behavior becomes more ragged at high temperatures and stronger deviations from the characteristic frequency are seen at 30{degree sign}, especially in figure 4C. The authors should discuss how these deviations impact their analysis of characteristic frequency.

Line 124 and 132: "suggests that CPGs may be involved…both time-scales are controlled by two nested levels of CPGs" these both seem like interpretations of the data and not things that are directly demonstrated by the data shown.

Figure 4A: It seems in this example that the relative durations of head grooming and leg rubbing, as well as the regularity of the behavior are altered at 30{degree sign}. The authors should address this. Behavioral confidence is also much lower under these conditions- is this an issue?

Line 248: "Our more surprising discovery was that the alternation between bouts of head cleaning and front leg rubbing is also periodic." I am not sure quite how surprising this is. I think it would equally surprising if there were no correlations in these measurements. I think it would help to have some measure of regularity/irregularity in these alterations.

*Reviewer #3:*

This study presents an intriguing example of rhythmic behavioral oscillations at a fast time scale embedded within a slower rhythmic behavioral oscillation and the similar effects of temperature on the rates of the two rhythms. The data and the writing are both exceptionally clear.

1) It is not clear why the authors use an unpublished example of different rates of oscillation in the first paragraph of the Introduction instead of using a published example, of which there are probably many to choose from.

2) At first, I was somewhat concerned that the temperature manipulation could not be used to assess the contribution of CPGs to the behaviors because temperature changes in these experiments affect both the CNS and the legs, the latter of which contain sensory neurons activated by leg movements. But then I realized that the "movement-related sensory feedback" for the slow rhythm would not be leg sensory feedback, but instead (possibly) the amount of dust on the head. I now think that generating constant bristle stimulation optogenetically indeed should make the likely sensory feedback for the slow rhythm constant, so the effect of temperature on the slow rhythm in this experiment does suggest a CPG for the slow rhythm.

3) Nonetheless, I think the first paragraph of the Discussion obscures rather than clarifies distinctions between effects on the CNS and PNS. It cites a series of studies that used localized cooling to implicate a CPG within a particular part of the CNS, as if the current experimental design is comparable to this, though in the current experiments the entire animal was warmed or cooled. It might be helpful to explicitly acknowledge that in the current study temperature was simultaneously manipulated in both the central and the peripheral nervous system, so the change in rhythm rate does not in itself eliminate the possibility that movement-related sensory feedback contributes to the rate, at least for the fast rhythm.

4) The sentence (lines 202-203), "These flies have no sensory stimuli except what they generate themselves by contact between their legs and bodies…" also seems misleading because it leaves out 1) any dust, etc. that happens to collect on their heads that was not put there by the experimenters and 2) leg proprioceptive input (i.e., not from contact).

5) Also, on line 241, the authors assert that the fast rhythm occurs "in the absence of patterned sensory input." This is misleading, because leg proprioceptors presumably provide patterned sensory feedback that could be used on this time scale. These statements should probably be revised to clarify.

[Editors’ note: further revisions were suggested prior to acceptance, as described below.]

Thank you for submitting your article "Behavioral evidence for nested central pattern generator control of *Drosophila* grooming" for consideration by *eLife*. Your article has been reviewed by 2 peer reviewers, and the evaluation has been overseen by a Reviewing Editor and K VijayRaghavan as the Senior Editor. The reviewers have opted to remain anonymous.

The reviewers have discussed their reviews with one another, and the Reviewing Editor has drafted this to help you prepare a revised submission. This is a strong paper now and the revisions are for enhancing clarity and presentation.

Essential revisions:

1) The apparent finding that the leg mechanosensory genetic ablation slows the slow rhythm, but not the fast rhythms, is intriguing, if true because it would indicate that the two rhythms can be independently modulated, which would suggest that they are independent CPGs, even though the slow one may trigger the fast one. (The differential effects of temperature on aspects of spontaneous grooming also suggest this.) But this conclusion rests on Figure 5D-E and H. Figure 5E shows what looks like no change in sweep frequency, but 5D shows what looks like a slowing of rub frequency; whether this apparent slowing is statistically significant or not is unclear. Figure 5H, for the slow rhythm, is a different kind of plot, so we have an "apples-to-oranges" comparison-there is no frequency power spectrum plot for the slow rhythm and no box plot of frequency for rubbing. So we are not certain that the slow rhythm slowed without the rub frequency also slowing. It would help to add the missing plots, allowing "apples-to-apples" comparisons, as well as statistical comparisons between control and tetanus toxin for each measure. This would be important for readers to be persuaded that the slow and fast rhythms can be independently modulated.

2) In this revised submission the authors introduce a new metric, Periodicity Index (PI) to quantify the periodicity of a time series (ratio of highest peak and most prominent shoulder). Although this seems intuitively logical it would help if there was a reference or (mathematical) justification for its use as a measure of periodicity.

3) Authors define a threshold (most prominent shoulder peak in ACF >0.2) for calling a behavior as periodic. This seems reasonable. But looking at the selected ACF samples for fast rhythms in Figure 1F and G, it seems like most of the prominent shoulder peaks fall below the 0.2 threshold. Unless we are mistaken, the authors seem to have chosen a particularly bad example to depict fast rhythms in Figure 1 of the paper.

4) The usage of P(f)-P(h) as explained by the authors in the rebuttal is understood; but not what the fly would be actually doing when this value approaches 0 (as happens a lot in Figures 4F and 6F). It would really help if the authors included sample videos corresponding to Figures 1D, 4F, and 6F.

*Reviewer #1:*

This study demonstrates that fruit flies have regular rhythms of head and leg grooming movements when dust is applied to the head and that they also alternate head and leg grooming with a regular, slower rhythm. These rhythms all increase with temperature and continue when head bristle stimulation is constant and when sensory feedback from the moving legs is reduced or eliminated. All of this suggests that the fly's central nervous system contains circuits-central pattern generators-for both the fast leg movements during grooming of either the head or the other leg and for the slow alternation between these two kinds of grooming. This suggests that central pattern generators can be nested within another central pattern generator at a higher hierarchical level, which is important for understanding how central nervous systems control movements generally.

from 2020: It is not clear why the authors use an unpublished example of different rates of oscillation in the first paragraph of the Introduction instead of using a published example, of which there are probably many to choose from.

Authors' response: We looked for an appropriate published example but came up blank. Crustacean antenna have historical contributions to ethology including grooming behavior, e.g.Vedel, J. P., 1982 and Barbato, J. C, 1997, and seemed an aesthetic example to illustrate how simple oscillators with different frequencies (and amplitudes) can create beautiful complex behaviors. This is anecdotal but reflects one author's fieldwork living next to the ocean. We can delete this example or exchange for a substitute if an alternative is proposed. Barbato, J. C., and Daniel, P. C. (1997). Chemosensory activation of an antennular grooming behavior in the spiny lobster, Panulirus argus, is tuned narrowly to L-glutamate. The Biological Bulletin, 193(2), 107-115. VEDEL, J. P. (1982). Reflex reversal resulting from active movements in the antenna of the rock lobster. Journal of Experimental Biology, 101(1), 121- 133.

My response to the authors' response:

FYI, some other examples of pairs of slow and fast CPG-driven (or apparently CPG-driven) rhythms that are coordinated across time scales (though this does not mean that one CPG triggers the other) are in:

Brown (1911) Quart J Exp Physiol 4:151-182 (rabbit hopping and scratching)

Carter and Smith (1986) J Neurophysiol 56:171-183 and 56:184-195 (cat walking and paw-shaking)

Weimann et al. (1991) J Neurophysiol 65:111-122 (crab gastric mill and pyloric rhythms)

Bartos et al. (1999) J Neurosci 19:6650-6660 (crab gastric mill and pyloric rhythms)

Esch et al. (2002) J Neurosci 22:11045-54 (leech crawling and swimming)

Zornik et al. (2010) J Neurophysiol 103: 3501-3515 (frog advertisement call sequence of fast and slow trills)

---

## [Author Response]

[Editors’ note: the authors resubmitted a revised version of the paper for consideration. What follows is the authors’ response to the first round of review.]

The reviewers agreed that the notion of nested CPGs for coordination of grooming is important but that stronger demonstration is required that the rhythms are indeed generated by a CPG in the absence of sensory feedback. In addition, better evidence that the two rhythms are coupled would significantly strengthen the study. Key experimental suggestions are below. Full comments of the reviewers are attached to provide further guidance.1) Better quantification of rhythms:a) High res leg tracking will help quantify both rhythms with more accuracy. It could also be used to better describe the coupling between the two rhythms (Figure 3 and 4, what are the "leg movements" that are invariant across temp).b) Poor confidence levels of the slow rhythm in Figure 4 and 5 need to be addressed with supporting videos, complementary analysis or new data from high res tracking.

We now confirm the existence of rhythms in the fast leg sweep/rub movements and in the slower alternation between bouts of head cleaning and front leg rubbing by analyzing new videos of grooming flies with higher spatial and temporal resolution. This required establishing a new behavior recording rig with a way to film the flies from below and implementation of Deep Lab Cut (DLC) software to track multiple leg joints. The new dataset includes recordings at 18º, 24º, and 30ºC; the results are included in the revised manuscript new Figure 1—figure supplement 4. We detect leg rub and sweep rhythms with frequencies increasing from ~5 Hz to 7 Hz, in agreement with our initial results as calculated from leg movement counts using the method described in new Figure 1—figure supplement 1.

Analysis using ABRS has sufficient temporal resolution to identify bouts of head sweeps and leg rubs, the time scale required to characterize the slower rhythm. The big advantage of ABRS is its ability to analyze large amounts of behavioral data to detect rhythms, patterns and correlations in noisy or variable signals, which was critical to detect the longer time-scale rhythm in the first place. We now confirm the periodicity of this long timescale bout alternation with the new videos analyzed using DLC in Figure 1—figure supplement 4D-F.

We quantify the periodicity of both short and long time-scale repeats using a new metric, the Periodicity Index, based on the auto-correlation function (Figure 1—figure supplement 2). The fact that there is periodicity at all on the long time-scale suggests a layer of behavioral organization, and the potential for an over-arching CPG, that is surprising. We acknowledge the greater variability and more ragged appearance of the longer-time-scale rhythm in optogenetically stimulated flies and include possible explanations in the revised Results and Discussion sections pertaining to these flies.

2) Role of proprioceptive feedback in rhythm generation:Proprioceptive feedback from the legs could be the major rhythmic input that drives both rhythms and its role should be explored by neuronal silencing (e.g. Mendes et al., 2013 eLife).

We address the potential role of proprioceptive feedback in rhythm generation by two additional experiments: single limb amputation and selective neuronal silencing. These approaches were inspired by Berendes et al. J Exp Bio 2016 and Mendes et al. *eLife* 2013, as suggested by Reviewer 1. Reducing or removing proprioceptive feedback by these experimental approaches slows the long time-scale alternation but does not eliminate the periodicity of either time-scale. These data are now included in the revised manuscript**.**

Grooming behavior is produced by a combination of internal pattern generating circuits initiated and modulated by responses to sensory cues. It is tricky to disentangle the relative contributions of these factors. The mechanosensory and proprioceptive cues that may contribute to grooming are also involved in other coordinated limb movements such as walking and postural control. Experiments presented in the original manuscript make the sensory experience more uniform using optogenetic activation. We now include new experiments that reduce sensory contributions by physical perturbations or genetic silencing. Together, these data allow us to estimate the contribution of this sensory information to rhythmic grooming behaviors (Figure 5, Figure 5—figure supplement 1).

Patterned proprioceptive input might be produced by and contribute to fast rhythmic leg movements. We reduced this feedback by amputating one front leg and tracking the stump using DLC. We find that movement of the truncated limb is still rhythmic on the short time-scale and very similar in frequency to that of intact legs, supporting the concept of CPG control (Figure 5—figure supplement 1). It was more difficult to assess the effect of amputation on the longer-time scale alternation because identification of head sweeps and leg rubs depends on detecting synchronization of the truncated limb with the intact one and their coordination is now less precise, as was previously reported for walking gaits (Berendes et al. 2016).

To block proprioceptive neuron activity, we used the same genetic combination as Mendes et al. 2013 (dacFlp, TubP>GAL80>; 5-40-GAL4, *UAS-TNT*) and a new split-GAL4 combination that targets leg proprioceptors (Figure 5B-C). These flies are somewhat uncoordinated but capable of producing rhythmic sweeps and rubs (Figure 5I). The long time-scale alternations are also rhythmic but slower than those of controls (Figure 5H), consistent with the slower rhythms observed in larval crawling in the absence of normal sensory feedback (Hughes and Thomas Mol Cell Neurosci 2007, Song et al. PNAS 2007, Pulver et al. J. Neurophys 2015).

Reviewer #1:Ravbar et al. have carried out behavioral analysis of *Drosophila* grooming behavior and have proposed that nested CPG modules are involved in generating the sub-routines during anterior grooming. It is a conceptually exciting hypothesis and is supported by authors' observation of two nested behavioral rhythms, a fast leg movement rhythm (5-7Hz) and a slower behavioral state switch rhythm (0.3-0.6Hz). The authors then use temperature change as a tool to prove existence and coupling of the underlying nested CPG modules that govern these rhythms.This work is conceptually interesting with a potential for broad interest, and I'm in favor of its publication provided the authors address the major points.1) The authors use temperature change as a tool to prove the existence and coupling of nested CPGs. Although there is a certain simplicity and elegance in this approach, it needs some more justification. Temperature directly influences neural activity (not restricted to CPGs). The authors cite important studies where temperature was used to pinpoint CPGs, but in these studies local heating/cooling of specific parts of the nervous system was used as a tool to increase/decrease neural activity in those areas and thereby locate the position of CPGs (e.g. Long and Fee 2008)). Global heating/cooling (as conducted in the current study) was only used in StG papers (Marder lab) to characterize the effect of temperature on previously well-described CPG networks. Given the pivotal role of these temperature change experiments in supporting central results of this paper, it needs more justification, both at a descriptive level as well as more characterization. E.g. it would be interesting to see how other attributes already tracked in these experiments (walking, hind grooming etc) are affected by temperature. Is there any behavior that is not proportionally affected by temperature and what does it tell us that is special about the anterior grooming rhythms? Also, given global heating will affect all neurons similarly, proportional changes in the two rhythms need not indicate "coupled" rhythms. This caveat should be discussed while discussing results from Figure 3.

Temperature does affect neural activity in general, potentially speeding up all behaviors. We expand our discussion of this possible explanation for the temperature-dependent increase in rhythm frequencies that we observe, but we do not think that the coupled increase in frequency of short and long time-scale rhythms is an epiphenomenon of generally faster neural activity because periodic behaviors are affected differently by temperature in spontaneous grooming. This caveat is now mentioned in the Results section Two time-scales contract together with temperature elevation and the counter-evidence described in section Nested CPGs can be decoupled in spontaneously grooming flies.

We reviewed the literature for fly behaviors but characterization of walking, courtship, or “fly-bowl” open-field data at multiple temperatures has not been reported to our knowledge. (This lack actually prompted us to initiate a collaboration to investigate the effects of temperature on courtship song.) Temperature dependent acceleration of oscillation has been used in several systems to provide evidence for the existence of Central Pattern Generator (CPG) circuits, as well as to investigate their anatomical locations, for example in frog vocalizations (Yamagouchi et al. 2008 and references within). This relationship between temperature and frequency prompts us to reason that behaviors that are triggered by acute sensory stimuli should be less affected by temperature than those governed by CPGs. In the revised manuscript we separate rhythmic from non-rhythmic behaviors on the long time-scale using a newly-devised Periodicity Index see Figure 1—figure supplement 2. We find that periodic long time-2 scale behaviors correlate better with temperature (R=0.83, p=0.02) than 2 non-periodic ones (R=0.70; p=0.08): new Figure 6—figure supplement 1. Taken together this new analysis shows that while all behaviors, acute and periodic, become faster with temperature, the latter show stronger temperature effect, as we would predict for CPG-driven behaviors.

Temperature-dependent acceleration of short and long time-scale rhythm frequency in grooming is one piece of evidence that these behaviors may be controlled by central pattern generator circuits, complementing our quantification of their periodicity and demonstration of their sensory independence.

2) A more direct evidence for CPGs would be to show that rhythmic motor output could be achieved without rhythmic sensory input and the authors also claim to have addressed this using optogenetics (Figure 4). The authors suggest that the changing quantity of dust on the fly's body as it grooms, could be a rhythmic input. This could be better explained. I'm guessing what matters is the difference in sensory input coming from head (H) versus front legs (F). If H>F then the fly does behavior "h", else it does "f". Doing behavior "h" reduces sensory input "H". With this assumption, the H-F input could indeed be rhythmic, and then the opto experiment rationale seems justified. However, for the fast rhythm (leg sweeps), the major rhythmic input is likely the proprioceptive input from the moving legs and not the continuously decreasing quantity of dust on the fly's body and hence this rhythm is not really addressed by the opto experiment. In fact the proprioceptive input from the legs could also be the defining input for the slower rhythm given that "number of leg movements" remain constant across temperatures (Figure 3 and 4K). One way to investigate pure centrally driven rhythms could be to eliminate rhythmic sensory feedback by leg amputation (e,g Berendes et al., J.Exp.Biol, 2016) or neuronal silencing (Mendes et al., eLife, 2013) and then apply dust/opto-stim and see if both fast and slow rhythms still persist.

The initial optogenetic stimulation experiments, activating mechanosensory bristle neurons over the body surface to mimic dust, were designed to eliminate the changing sensory drive that occurs as dust is removed from anterior body parts during sustained grooming, flattening the normal anterior-to-posterior grooming progression. This optogenetic stimulus likely also masks potential sensory contributions to the long time-scale rhythmic alternation of bouts of head-sweeps and front-leg rubs from acute evaluation of dust load on the legs. The reviewer is correct that mechanosensory bristle activation by Chrimson does not eliminate the proprioceptive and self-generated sensory stimulation produced by the movement of the legs or their contact with the body. If this contact is rhythmic, it could contribute to maintaining the head sweep or leg rub rhythms on the short time-scale.

As the reviewer suggests, we now use alternative methods to reduce sensory feedback, assessing short and long time-scale periodicity in flies with an amputated leg or genetically silenced proprioceptive sensory neurons. We analyzed grooming rhythms using both ABRS and limb tracking with Deep Lab Cut. The results are described above in the response to the Editors’ summary, but in brief, we see that the short timescale movements remain rhythmic, and the head sweep/leg rub cycles also remain periodic but less so, and become much slower than those in the control group. These data are now included in a new Figure 5 and Figure 5 —figure supplement 1.

We also observe and quantify a reduction in the amount of periodic behavior in the optogenetically stimulated flies, especially at higher temperatures (Figure 1—figure supplement 4H), suggesting the possibility that appropriate sensory feedback supports the stability of the rhythms. These results suggest that while the basic rhythms may be CPG generated, sensory information contributes to speed and stability of both time-scales. We now mention this possibility in an expanded in the Results section Two time-scales contract together with temperature elevation and in the Discussion where we provide examples from other systems.

3) Finally, the authors describe an interesting "decoupling" of the two rhythms during spontaneous grooming. The authors observe similar leg movement frequencies, however the longer f-f and h-h rhythms seem very rare and variable. So the "decoupling" between the two rhythms is in fact driven by "absence" of the slower rhythm in the experimental conditions. This is also interesting but should be discussed appropriately. Looking at the full ethogram in Figure 5B, it is apparent that as temperature increases, flies spend more time walking versus grooming and so the grooming bouts are likely too short to show any slow rhythms. So in theory, the rhythms could be still coupled, just now more disrupted because of a competing "walking" state.

The f-f and h-h oscillations are indeed less frequent in spontaneous grooming. They are less robust at room temperature, and they become increasingly “ragged” at high temperature, when flies do spend more time walking. This does make it harder to get a large enough sample to analyze the periodicity of the long time-scale alternation or whether its rate increase is coupled that of the short time-scale rhythms as temperature increases.

But when spontaneous grooming does occur, its long time-scale frequency does increase, and not at the same rate as the short time-scale frequency. The fact that we observe this lesser but significant increase in ff- and hh-cycle oscillations in these flies suggests that the low sample size is not the only explanation for the “decoupling.” We propose that the short time-scale CPG is always ON (during grooming) whereas the long timescale CPG can be either ON or OFF. We show the overall lower amount of periodic long time-scale behavior in spontaneously grooming flies across all temperatures in a newly added Figure 1—figure supplement 3G and an example of the switching between periodic and non-periodic behaviors in these flies is shown in Figure 6—figure supplement 2B. As we described above, we can now separate the periodic from non-periodic behaviors (on the long time-scale) in these flies (Figure 6—figure supplement 1). On average the ff- and hh-cycles in spontaneously grooming flies do not scale with temperature to the same extent as in the dust-stimulated flies; however, if we only sample from the periodic behaviors (CPG is ON), the temperature correlation is closer to that of the 2 dust-stimulated flies (R=0.83, p<0.02) while non-periodic behaviors 2 correlate with temperature less significantly (R=0.70, p<0.07) (Figure 6— figure supplement 1). We interpret that as flies alternating between periodic CPG-driven and non-periodic behaviors.

The competition between grooming and walking is an interesting future direction for research: perhaps the two rhythmic leg movement-based behaviors share CPG components.

4) The leg movement analysis needs to be justified more than what it is in the Methods section. The authors acquire videos of grooming flies at 60 Hz,then select frames containing anterior grooming bouts and in these selected frames, they then use pixel difference in consecutive frames and use the frames depicting the largest movement for quantifying frequency. It is unclear what aspect of leg movement the authors are quantifying in this analysis and whether it is affected by video frame-rate or camera angle. What motor action is represented by the "leg movement" frequency? Why is this frequency value higher for "f" bouts versus "h" bouts? Would the authors get the same results if videos were captured at a higher frame rate? It is important to disambiguate this, because the authors use "number of leg movements" as an attribute in Figure 3 to prove coupling between the CPGs. The authors could acquire some additional videos, preferably at a high frame rate, and use one of the recent tools (Pereira et al., 2019, Mathis et al., 2018) to track the legs during anterior grooming, as has been done for leg movement rhythm quantification during walking. If this type of high resolution analysis also suggests a similar leg sweep frequency, the current analysis framework of the authors will be justified. Moreover, it could provide a better understanding of results in Figure 3 and 4K.

We have expanded our description of how we initially analyzed leg movements in the Methods section and justify why the 60Hz frame rate is sufficient to count leg sweeps and rubs (Figure 1 —figure supplement 1), but to address the core concern, we collected new data with a different camera that enables high resolution limb tracking using DeepLabCut (DLC) at 100Hz frame rate as suggested. This analysis detects leg movement frequencies of ~5-7Hz at temperatures from 18-30ºC, which is in good agreement with our previous measurements. This new analysis also confirms the periodicity of both the short and the long time-scales of grooming behaviors. These results are now shown in Figure 1—figure supplement 4.

The frequency of f-bouts appears slightly higher than that of h-bouts in the original analysis (Figure 2—figure supplement 1) although we couldn’t detect a significant difference between them with the DLC analysis. This may be because the legs have to travel a greater distance during a head sweep than during a leg rub, or because head sweeps are differently affected by sensory feedback, as suggested by (Figure 5D-E). Head sweeps and leg rubs are likely controlled by the same CPG but modulated differently by their sensory feedback, a possibility we plan to explore in future experiments.

5) For the slow f-f and h-h rhythm quantification the authors use an interesting approach (P(f)-P(h)). Although this looks promising in Figures 1D, 2F and supp Figure 2, many of the peak/trough values tend to be very close to zero in Figures 4F, 5F making it unclear what rhythms does this really represent in terms of actual behavior. Are we still talking about clear rhythms between 2 states as depicted in Figure 1A? Performing oscillation analysis using confidence values between artificially defined binary states can enforce the output to look like an oscillation and hence it is important to explicitly discuss this.

We expand the description of the approach in the Methods section. As the reviewer said, we identify the two states (h- and f-bouts) by their probability (our detection confidence). To measure the frequency of bout alternations, we use the time distance between local confidence minima and maxima of P(f) – P(h) (the bout centroids shown in Figure 1D-E) rather than bout durations because the former is less error-prone than the positions of onsets and offsets (edges) of bouts. Therefore, our measurement is not affected by our definition of the binary states.

Reviewer #2:This manuscript uses temperature and optogenetic manipulations to alter the speed of grooming behaviors and examines the resulting changes in fast- and slow-time scale behavioral structure. The authors find that temperature increases the frequency both of individual leg movements, and the alteration between bouts leg rubbing and head grooming, such that the number of leg cycles per bout is approximately the same across a range of temperatures. A similar results is found when grooming is elicited by optogenetic stimulation of mechanosensory bristles, rather than by dust. Finally, the authors show that coordination between these two levels of behavioral organization breaks down during spontaneous grooming. The authors draw conclusions about the control of these behaviors by nested central patterns generators. The behavioral observations presented here are interesting but I felt that more analysis of genetic or circuit manipulations would be required to firmly draw the conclusions the authors would like to about the neural control of these behaviors.A central claim of the manuscript is that *Drosophila* grooming may be controlled by nested CPGs. As currently presented the evidence is not especially strong, as it is not clear what the alternative to central pattern generator control of these behaviors would be. The manuscript would be stronger if the authors identified neurons whose activation or inactivation altered behavioral rhythms or the coupling between fast and slow-time scale structure in the behavior.In the case of high temperature + constant sensory stimulation the behaviors seem much more variable and irregular but this is not noted or discussed.

The longer-time-scale rhythms, alternating bouts of head sweeps and front leg rubbing, are indeed more variable at higher temperature and under optogenetic stimulation. This is now quantified in the new Figure 1— figure supplement 3 where we show the amounts of periodic behaviors on the long time-scale across the temperatures and for different datasets, including the optogenetically stimulated flies. We also observe this effect in the new data where we genetically reduce proprioceptive inputs as shown in the new Figure 5. We developed a metric to quantify the amount of periodicity in all genotypes and conditions, as described in Figure 1 —figure supplement 2, and we include possible explanations in the Results as well as in a new paragraph in the Discussion section. For the optogenetically-stimulated flies, in particular, we speculate that normal sensory feedback may be dispensable for basic rhythmicity but contribute to appropriate timing and stability, as has been reported in other systems as we now mention the Discussion.

Line 35-36: "Automating the sequence by calling it actions in series produces reliable control" This seems like a bit of straw man. Most rhythmic behaviors such as walking, breathing, etc, are assumed or known to arise from central pattern generators, and not from individual cycle-by-cycle decision, although of course each behavior can be modulated on a cycle-by-cycle basis.

We did start from the assumption that the short time-scale rhythm of individual leg movements for grooming would be CPG generated, but this had yet to be formally demonstrated. Our discovery that the alternation between bouts, the longer time-scale, was also rhythmic was a genuine surprise.

Line 73: "characteristic frequency of 6Hz." It seems in 1C that the characteristic frequency of head grooming and leg rubbing are slightly different. Can those averages be shown or given here and are they significantly different?

Head sweeps are slightly slower than leg rubs. We quantify this in Figure 2—figure supplement 1. The frequencies are corroborated using DeepLabCut on new video as shown in Figure 1 —figure supplement 4.

Line 125 + figure 2C/4C. The authors note that the behavior becomes more ragged at high temperatures and stronger deviations from the characteristic frequency are seen at 30{degree sign}, especially in figure 4C. The authors should discuss how these deviations impact their analysis of characteristic frequency.

We have quantified the periodicity of the longer-time scale under various conditions and include a discussion of the potential role of sensory feedback in maintaining more regular rhythmicity as discussed in the response to major comments above.

Using the new quantification, we can see a sharp decrease in periodicity index and in the proportion of periodic behavior at 30º in the optogenetically stimulated flies (Figure 1—figure supplement 3E and H). We believe that some sensory feedback may be necessary to stabilize the rhythmic output of the CPGs, since this is consistent with our new data reducing sensory feedback, and we now mention this possibility in the Discussion.

Line 124 and 132: "suggests that CPGs may be involved…both time-scales are controlled by two nested levels of CPGs" these both seem like interpretations of the data and not things that are directly demonstrated by the data shown.

The core proposal of the manuscript is that nested CPGs are the model that best explains our data. We have reviewed the division of Results and Discussion carefully to separate descriptions of behavior from interpretation. Since the hypothesis that CPGs may contribute to grooming guides our experiments, we do mention this in the experimental design, but we have tried to reserve interpretation for the Discussion.

Figure 4A: It seems in this example that the relative durations of head grooming and leg rubbing, as well as the regularity of the behavior are altered at 30{degree sign}. The authors should address this. Behavioral confidence is also much lower under these conditions- is this an issue?

The duration of individual leg movements and the lengths of bouts are reduced at higher temperature. The rhythmicity of leg movements remains strong but the alternation between bouts does become less precise. We now quantify the strength and the amount of periodicity in Figure 1— figure supplement 3 E and H. The overall result, that there is temperature-dependent contraction of the long time-scale, remains robust.

Line 248: "Our more surprising discovery was that the alternation between bouts of head cleaning and front leg rubbing is also periodic." I am not sure quite how surprising this is. I think it would equally surprising if there were no correlations in these measurements. I think it would help to have some measure of regularity/irregularity in these alterations.

We expected periodicity of leg rubbing and head sweeps; these highly repetitive movements seemed characteristic of CPG control. Our previous work revealed that both behavior identity and duration affect the choice of next movement (Mueller et al. 2019), but prior to this study, we had not identified rhythmic alternation between anterior grooming bouts.

The long time-scale alternation is more variable that the short time-scale, especially at higher temperatures, under optogenetic stimulation, or in the absence of sensory feedback. We now quantify the regularity/irregularity of these alternations by the Periodicity Index, described in Figure 1 —figure supplement 2. We can also separate periodic from non-periodic alternations and estimate the overall amount of periodic behavior in different conditions (Figure 1 —figure supplement 3H).

Reviewer #3:This study presents an intriguing example of rhythmic behavioral oscillations at a fast time scale embedded within a slower rhythmic behavioral oscillation and the similar effects of temperature on the rates of the two rhythms. The data and the writing are both exceptionally clear.1) It is not clear why the authors use an unpublished example of different rates of oscillation in the first paragraph of the Introduction instead of using a published example, of which there are probably many to choose from.

We looked for an appropriate published example but came up blank. Crustacean antenna have historical contributions to ethology including grooming behavior, e.g.Vedel, J. P., 1982 and Barbato, J. C, 1997, and seemed an aesthetic example to illustrate how simple oscillators with different frequencies (and amplitudes) can create beautiful, complex behaviors. This is anecdotal but reflects one author’s fieldwork living next to the ocean. We can delete this example or exchange for substitute if an alternative is proposed.

Barbato, J. C., and Daniel, P. C. (1997). Chemosensory activation of an antennular grooming behavior in the spiny lobster, Panulirus argus, is tuned narrowly to L-glutamate. The Biological Bulletin, 193(2), 107-115.

Vedel, J. P. (1982). Reflex reversal resulting from active movements in the antenna of the rock lobster. Journal of Experimental Biology, 101(1), 121133.

2) At first, I was somewhat concerned that the temperature manipulation could not be used to assess the contribution of CPGs to the behaviors because temperature changes in these experiments affect both the CNS and the legs, the latter of which contain sensory neurons activated by leg movements. But then I realized that the "movement-related sensory feedback" for the slow rhythm would not be leg sensory feedback, but instead (possibly) the amount of dust on the head. I now think that generating constant bristle stimulation optogenetically indeed should make the likely sensory feedback for the slow rhythm constant, so the effect of temperature on the slow rhythm in this experiment does suggest a CPG for the slow rhythm.

This was also our reasoning on the optogenetic stimulation of mechanosensory bristles. It should make the competing sensory drives between body parts more uniform and flatten out external sensory contributions to the periodicity, revealing possible CPG control, accelerated by temperature. We clarified our description of the goals of the optogenetic activation in the corresponding Results section, Periodicity and correlation between time-scales persist when sensory stimulation is constant.

3) Nonetheless, I think the first paragraph of the Discussion obscures rather than clarifies distinctions between effects on the CNS and PNS. It cites a series of studies that used localized cooling to implicate a CPG within a particular part of the CNS, as if the current experimental design is comparable to this, though in the current experiments the entire animal was warmed or cooled. It might be helpful to explicitly acknowledge that in the current study temperature was simultaneously manipulated in both the central and the peripheral nervous system, so the change in rhythm rate does not in itself eliminate the possibility that movement-related sensory feedback contributes to the rate, at least for the fast rhythm.

We have modified the Discussion according to this suggestion: “Although here we change the temperature of the whole fly…” We are indeed changing the temperature of the whole animal to argue for internal CPG control of rhythmicity over two time-scales. We now also perform new experiments that disrupt proprioceptive sensory feedback to address possible contributions from the PNS (Figure 5 and Figure 5—figure supplement 1).

4) The sentence (lines 202-203), "These flies have no sensory stimuli except what they generate themselves by contact between their legs and bodies…" also seems misleading because it leaves out 1) any dust, etc. that happens to collect on their heads that was not put there by the experimenters and 2) leg proprioceptive input (i.e., not from contact).

In the revised manuscript section on spontaneously grooming flies, we rewrote this statement as follows: “These flies have no experimentally applied sensory stimuli - only what they generate themselves by contact between their legs and bodies, and the associated proprioceptive feedback - so these motor patterns are most likely to be generated by internal circuits.” We hope this acknowledges the limits of our control: there may be external or internal stimuli the flies sense that the experimenters did not intend. (We address the proprioceptive input below.)

5) Also, on line 241, the authors assert that the fast rhythm occurs "in the absence of patterned sensory input." This is misleading, because leg proprioceptors presumably provide patterned sensory feedback that could be used on this time scale. These statements should probably be revised to clarify.

In the revised manuscript, we attempt to address potential proprioceptive contributions through additional amputation and genetic silencing experiments, finding that proprioceptive feedback is not essential for short time-scale rhythmicity but makes long time-scale rhythms slower (Figure 5). Even with an amputated leg the frequency of the stump is nearly identical to that of the intact leg (Figure 5—figure supplement 1).

We retained this statement in the Discussion: “Here we investigate what aspects of fly grooming are periodic, demonstrating that the short timescale leg sweeps and rubs repeat at characteristic frequencies, in the absence of patterned sensory input, and in a temperature-dependent manner (Figure 2D-E), consistent with control by central pattern generators.” We hope that the optogenetic activation, amputation, and genetic silence experiments, as well as the spontaneous grooming analysis, now provide enough evidence to support this assertion.

[Editors’ note: what follows is the authors’ response to the second round of review.]

Essential revisions:1) The apparent finding that the leg mechanosensory genetic ablation slows the slow rhythm, but not the fast rhythms, is intriguing, if true because it would indicate that the two rhythms can be independently modulated, which would suggest that they are independent CPGs, even though the slow one may trigger the fast one. (The differential effects of temperature on aspects of spontaneous grooming also suggest this.) But this conclusion rests on Figure 5D-E and H. Figure 5E shows what looks like no change in sweep frequency, but 5D shows what looks like a slowing of rub frequency; whether this apparent slowing is statistically significant or not is unclear. Figure 5H, for the slow rhythm, is a different kind of plot, so we have an "apples-to-oranges" comparison-there is no frequency power spectrum plot for the slow rhythm and no box plot of frequency for rubbing. So we are not certain that the slow rhythm slowed without the rub frequency also slowing. It would help to add the missing plots, allowing "apples-to-apples" comparisons, as well as statistical comparisons between control and tetanus toxin for each measure. This would be important for readers to be persuaded that the slow and fast rhythms can be independently modulated.

We now use the same type of plotting (the box plots) for all the measures in Figure 5 (D-K). We also added the missing statistics (p-values) to each panel. The slowing of the rub frequency shown in Figure 5D is statistically significant (p < 0.001) as is the slowing down of the long time-scale (p = 0.003) shown if Figure 5H.

2) In this revised submission the authors introduce a new metric, Periodicity Index (PI) to quantify the periodicity of a time series (ratio of highest peak and most prominent shoulder). Although this seems intuitively logical it would help if there was a reference or (mathematical) justification for its use as a measure of periodicity.

We explored metrics that could be used to quantify periodicity but did not find a more formal, mathematical solution. The ACF itself, of course, can be used to detect cycles in the data. We use the π as a threshold to determine which behaviors are periodic, rather than as an optimal estimate of the strength of periodicity.

3) Authors define a threshold (most prominent shoulder peak in ACF >0.2) for calling a behavior as periodic. This seems reasonable. But looking at the selected ACF samples for fast rhythms in Figure 1F and G, it seems like most of the prominent shoulder peaks fall below the 0.2 threshold. Unless we are mistaken, the authors seem to have chosen a particularly bad example to depict fast rhythms in Figure 1 of the paper.

Thank you for spotting this. We had plotted slightly smoothed (to remove high frequency noise) examples of ACFs but had not normalized them afterwards (the autocorrelation at t=0 should be 1). We revised the figures that show the ACFs accordingly and this does not affect the π measurements.

4) The usage of P(f)-P(h) as explained by the authors in the rebuttal is understood; but not what the fly would be actually doing when this value approaches 0 (as happens a lot in Figures 4F and 6F). It would really help if the authors included sample videos corresponding to Figures 1D, 4F, and 6F.

We now include example videos of the critical transitions. When the value approaches 0, the flies are switching between leg rubbing and head sweeping behaviors and ABRS does not assign the spatiotemporal feature to either category.

In response to this critique, we carefully reviewed the raw videos of the flies from Figures 1D, 4F, and 6F and concluded that the threshold for which events should be assigned to either f- or h- peaks (or valleys) was set too liberally. We adjusted the threshold so that the absolute value should be at least 0.05 and recomputed the ff/hh cycles. This change affected the long time-scale analysis in Figures 1D, 4F, and 6F, resulting in smaller sample sizes. We now combine ff and hh cycles into a single value, and repeated our analysis of the temperature dependence of the long-time scale. The core conclusions remain the same, with the exception of Figure 6 Supplement 1, where the infrequent long time-scale periodic behaviors no longer show a temperature effect.

Reviewer #1:This study demonstrates that fruit flies have regular rhythms of head and leg grooming movements when dust is applied to the head and that they also alternate head and leg grooming with a regular, slower rhythm. These rhythms all increase with temperature and continue when head bristle stimulation is constant and when sensory feedback from the moving legs is reduced or eliminated. All of this suggests that the fly's central nervous system contains circuits-central pattern generators-for both the fast leg movements during grooming of either the head or the other leg and for the slow alternation between these two kinds of grooming. This suggests that central pattern generators can be nested within another central pattern generator at a higher hierarchical level, which is important for understanding how central nervous systems control movements generally.from 2020: It is not clear why the authors use an unpublished example of different rates of oscillation in the first paragraph of the Introduction instead of using a published example, of which there are probably many to choose from.Authors' response: We looked for an appropriate published example but came up blank. Crustacean antenna have historical contributions to ethology including grooming behavior, e.g.Vedel, J. P., 1982 and Barbato, J. C, 1997, and seemed an aesthetic example to illustrate how simple oscillators with different frequencies (and amplitudes) can create beautiful complex behaviors. This is anecdotal but reflects one author's fieldwork living next to the ocean. We can delete this example or exchange for a substitute if an alternative is proposed. Barbato, J. C., and Daniel, P. C. (1997). Chemosensory activation of an antennular grooming behavior in the spiny lobster, Panulirus argus, is tuned narrowly to L-glutamate. The Biological Bulletin, 193(2), 107-115. VEDEL, J. P. (1982). Reflex reversal resulting from active movements in the antenna of the rock lobster. Journal of Experimental Biology, 101(1), 121- 133.My response to the authors' response:FYI, some other examples of pairs of slow and fast CPG-driven (or apparently CPG-driven) rhythms that are coordinated across time scales (though this does not mean that one CPG triggers the other) are in:Brown (1911) Quart J Exp Physiol 4:151-182 (rabbit hopping and scratching)Carter and Smith (1986) J Neurophysiol 56:171-183 and 56:184-195 (cat walking and paw-shaking)Weimann et al. (1991) J Neurophysiol 65:111-122 (crab gastric mill and pyloric rhythms)Bartos et al. (1999) J Neurosci 19:6650-6660 (crab gastric mill and pyloric rhythms)Esch et al. (2002) J Neurosci 22:11045-54 (leech crawling and swimming)Zornik et al. (2010) J Neurophysiol 103: 3501-3515 (frog advertisement call sequence of fast and slow trills)

We chose to keep the lobster example but have made it clear that it is an inspirational observation that motivated our study; we add a description of a piano player to illustrate the general question that fascinates us. We thank the reviewer for pointing out these relevant references, some of which we now include in the Discussion.